

**TITLE**
**Seismic survey in urban area: the activities of the EMERSITO INGV**
**emergency group in Ancona (Italy) following the 2022 $M_W$ 5.5 Costa**
**Marchigiana-Pesarese earthquake**
**Authors**: Daniela Famiani (1), Fabrizio Cara (1), Giuseppe Di Giulio (2), Giovanna Cultrera
(1), Francesca Pacor (3), Sara Lovati (3), Gaetano Riccio (4), Maurizio Vassallo (2), Giulio
Brunelli (3), Antonio Costanzo (11), Antonella Bobbio (5), Marta Pischiutta (8), Rodolfo
Puglia (3), Marco Massa (3), Rocco Cogliano (4), Salomon Hailemikael (1), Alessia Mercuri
(1), Giuliano Milana (1), Luca Minarelli (2), Alessandro Di Filippo (5), Lucia Nardone (5),
Simone Marzorati (10), Chiara Ladina (10), Debora Pantaleo (10), Carlo Calamita (10), Maria
Grazia Ciaccio (1), Antonio Fodarella (4), Stefania Pucillo (4), Giuliana Mele (1), Carla Bottari
(6), Gaetano De Luca (7), Luigi Falco (4), Antonino Memmolo (4), Giulia Sgattoni (9),
Gabriele Tarabusi (9)
Affiliation:
(1) Istituto Nazionale di Geofisica e Vulcanologia, Sezione di Roma1, Roma, Italy.
(2) Istituto Nazionale di Geofisica e Vulcanologia, Sezione di Roma1, L'Aquila, Italy.
(3) Istituto Nazionale di Geofisica e Vulcanologia, Sezione di Milano, Milano, Italy.
(4) Istituto Nazionale di Geofisica e Vulcanologia, Sezione Irpinia, Grottaminarda, Italy.
(5) Istituto Nazionale di Geofisica e Vulcanologia, Sezione Osservatorio Vesuviano,
22       Napoli, Italy.
(6) Istituto Nazionale di Geofisica e Vulcanologia, Sezione Osservatorio Etneo, Catania,
24       Italy.
(7) Istituto Nazionale di Geofisica e Vulcanologia, Sezione Osservatorio Nazionale
26       Terremoti, L'Aquila, Italy.
(8) Istituto Nazionale di Geofisica e Vulcanologia, Sezione di Roma2, Roma, Italy.
(9) Istituto Nazionale di Geofisica e Vulcanologia, Sezione di Bologna, Bologna, Italy.
(10) Istituto Nazionale di Geofisica e Vulcanologia, Sezione Osservatorio Nazionale
Terremoti, Ancona, Italy.
(11) Istituto Nazionale di Geofisica e Vulcanologia, Sezione Osservatorio Nazionale
Terremoti, Rende, Italy.
Correspondence to:
Fabrizio Cara fabrizio.cara@ingv.it
Daniela Famiani daniela.famiani@ingv.it
**Abstract**
This paper illustrates the activities of EMERSITO, an emergency task force of the *Istituto*
*Nazionale di Geofisica e Vulcanologia* (INGV, Italy) devoted to site effects and microzonation
studies, during the seismic sequence that occurred close to the Adriatic coast in Central Italy
since November 9th, 2022, following the Mw 5.5 mainshock localised in the sea. In particular,
we describe the steps that led to the deployment of a temporary network of seismic stations in
the urban area of Ancona, the main city of the Adriatic coastline. Data collected by the
temporary Ancona network (identification code 6N, doi: 10.13127/sd/qctgd6c-3a, EMERSITO
Working Group, 2024) from November 2022 to the end of February 2023 have been



preliminary analysed with different techniques to characterise the deployment sites, and are
now available for further and detailed studies.

## 1. Introduction

On November 9th, 2022, at 06:07:24 UTC (07:07:24 local time), a $M_W$ 5.5 earthquake localised
in the Adriatic Sea struck the Marchigiana-Pesarese coast in Central Italy (Fig. 1). Due to its
magnitude, exceeding the threshold of 5.0, and the closeness to urban areas (Fano and Pesaro
are about 30-35 km, Ancona 45 km far from the epicenter), *Istituto Nazionale di Geofisica e*
*Vulcanologia* (National Institute of Geophysics and Volcanology, INGV[1]) soon activated the
Seismic Crisis Unit to monitor the ongoing seismic sequence. Among several tasks, the Crisis
Unit coordinates the INGV emergency task forces[2] devoted to specific issues and scientific
support for the activities of the Civil Protection: SISMIKO[3] (Moretti et al. 2023), for adding
seismic stations in the epicentral area to improve the localization of the seismic events of the
sequence, EMERGEO[4] for investigating the surface geological effects, QUEST[5] for the
macroseismic survey and EMERSITO[6] for site effects and seismic microzonation studies. In
general, the INGV task forces[2] operate synergistically although with a different intervention
timing. In particular, SISMIKO[3], EMERGEO[4] and QUEST[5] start their activities within a few
hours to 1-2 days after the mainshock. EMERSITO[6] activities, on the contrary, usually start
from 2 to 7 days after the main seismic event, depending on the level of damage caused by the
mainshock and, therefore, the accessibility to the epicentral area where the site effect are often
more evident (Cara et al. 2019).
In this paper, we focus on the activities of EMERSITO[6] working group following the $M_W$ 5.5
mainshock in the Adriatic sea. The area of the Adriatic coast where the earthquake was felt was
very broad, approximately ranging from the cities of Rimini and Ancona that are about 90 km
far from each other (Fig. 1). However, the level of damage, reported by both the fire brigade
and the QUEST[5] surveys, was very low (maximum IV MCS), so the logistics left us some
options to plan an intervention for site effects studies. After several considerations,
EMERSITO[6] decided to deploy a temporary seismic network in the urban area of Ancona, the
regional capital of Marche. This choice was driven by: a) the relative high values of peak
ground acceleration (PGA) recorded for the mainshock (the maximum PGA has been recorded
in Ancona at IV.PCRO station with 197 cm/s$^2$ on the EW component); b) the damage and
evacuations reported by the fire brigade and the technicians of Marche region; c) the strong
lithological heterogeneities in town; d) the scientific interest in improving the approach for the
evaluation of the local seismic response in urban areas.
The deployment of the network started 4 days after the mainshock and was completed in three
days, also taking advantage of the presence of an INGV office in Ancona[7] and with the
collaboration of the municipality and of the Marche Region technicians. During the emergency,
which lasted from November 2022 to March 2023, EMERSITO[6] carried out four public reports
to describe its activities (Cara et al., 2022a, 2022b, 2022c; Famiani et al., 2023).
In this paper we describe in detail the EMERSITO[6] network, the data collected and some
preliminary analyses.

## 2. Deployment of the temporary network

### 2.1 Seismological and geological framework



The 2022 $M_W$ 5.5 seismic sequence struck the Adriatic coast and affected some major towns,
such as Pesaro, Rimini, Fano, Senigallia and Ancona among others (Fig. 1). This latter city
(about 100.000 citizens) is the administrative center of the Marche region and one of the main
seaports of the Adriatic Sea. Before this event, in the previous century Ancona was hit by
significant earthquakes: in 1930 (epicenter close to Senigallia city, 10-15 km far from Ancona,
estimated Mw 5.8 and MCS intensity VIII; Guidoboni et al. 2018, Rovida et al. 2020 and 2022;
see Fig. 1) and more recently in 1972 by an important seismic sequence (Kissilinger 1972,
Console et al. 1973) that lasted 11 months. The shocks of the 1972 sequence were short in
duration but showed rather high values of PGA; the strongest earthquake occurred on June 14,
with magnitude $M_W$ 4.7 and estimated MCS intensity VIII. The epicenter of this event was
localized in the Adriatic sea in front of the Ancona seaport (Fig. 1), at about 10 km from Ancona
downtown in the NE direction (Rovida et al, 2017). The city experienced diffuse but moderate
damage with 7000 of 35000 buildings declared unusable. More than 30.000 people left their
homes. At the end of the 1972 sequence, Ancona was the object of the first large-scale seismic
monitoring in Italy, with the deployment of a network (Ferraris et al., 1975) followed by an
extensive microzonation survey of the area (Calza et al., 1981). The reconstruction, also in
downtown, was exemplary for the Italian standards and followed strict anti-seismic rules.
During the 2022 mainshock, localized at a distance of about 45 km from Ancona (see Fig. 1),
the city experienced some negligible damage and evacuations, as reported by the regional
technicians and the Fire Brigade (Fig. 2). As for the 1972 event, higher levels of PGA were
recorded during the main shock compared with instrumented sites at similar distance
(Engineering Strong Motion Database-ESM[8], Luzi et al., 2020). A subset of the recorded PGA
values are reported in Table 1 (see also Figure 1 for details in the position of the considered
instrumented sites).
**Table 1**: PGA recorded by some stations of the two permanent networks in Italy, IV
(https://doi.org/10.13127/SD/X0FXnH7QfY) and IT (https://doi.org/10.7914/SN/IT), ordered by epicentral
distance. The two stations in Ancona are highlighted in bold.

| Network | Station | Locality | Epicentral distance (km) | Horizontal PGA (cm/s$^2$) |
|---|---|---|---|---|
| IV | COR1 | Corinaldo | 49.3 | 31.610 |
| **IT** | **ANB** | **Ancona** | **48.8** | **166.424** |
| IV | FCOR | Fonte Corniale | 48.6 | 21.796 |
| **IV** | **PCRO** | **Ancona** | **47.9** | **197.842** |
| IT | CTL | Cattolica | 47.3 | 31.749 |
| IV | CRTC | Cartoceto | 44.2 | 22.409 |
| IV | SENI | Senigallia | 34.6 | 139.209 |
| IV | FANO | Fano | 30.5 | 52.613 |

From a geological point of view, Ancona is characterized by strong lithological heterogeneity
and represents a scientifically interesting case for the evaluation of the local seismic response
in an urban area. Moreover, the western area of Ancona is built on a deep landslide (Stucchi et
al., 2005; Stucchi and Mazzotti, 2009). In 1982, after a period of heavy rain, the landslide
moved suddenly (Crescenti et al., 2005), involving several suburban districts of Ancona:



Posatora, Borghetto and partially Torrette (Fig. 3). The movement of the landslide damaged
two hospitals and the Faculty of Medicine of the University, 280 buildings were destroyed and
overall 865 homes damaged, the railway was torn up and the coastal road was damaged along
a front of approximately 2.5 kilometers. The disaster forced the authorities to evacuate 3,661
people from the affected area. Nowadays the landslide zone, as well the aquifer, is constantly
monitored through an early-warning system (Cardellini and Osimani, 2013) and it is still in
very slow movement (Agostini et al., 2014).
The Ancona area falls in the marginal part of the central Apennines thrust system, where Mio-
Plio-Pleistocene terrigenous deposits overlie a mostly carbonate succession referable to the
Umbria-Marche succession (Cello and Tondi, 2013). In the periadriatic sector, the geological
structures related to the origin of the central Apennine chain are generally buried under the
foredeep turbidite successions that sedimented starting from the Miocene age (Bally et al.,
1986). In particular, in the area of Ancona (Fig. 4), this foredeep succession is mainly
characterized, in its upper part, by Pleistocene gray-blue marly clays (*Argille Azzurre*, FAA
formation). During the Late Pliocene there was an intense phase of regional uplift that in the
Middle Pleistocene, resulted in the emergence of the external part of the Marche region from
the sea level. Subsequently, and in relation to the different climatic phases, there were erosion
processes of various intensity (also stasis), and sedimentation. All these phenomena modeled
the landscape defining the current morphostructural arrangement of the region and producing
alluvial, eluvial-colluvial marine and landscape deposits widely outcropping in the study area.
The recent anthropization and urbanization are strongly altering the original morphology, in
particular in the coastal area, introducing erosion and accumulation processes that are
considerably more rapid and intense than those due to natural causes (Farabollini et al. 2000).
The outcropping marine succession in Ancona has been classified into four lithostratigraphic
units from bottom to top:
a) *Schlier* formation (SCH)
b) Chalky-sulfur formation (GES)
c) *Colombacci* formation (FCO)
d) *Argille azzurre* formation (FAA)
SCH formation (Late Miocene age, hemipelagic origin) diffusely outcrops along the coastline
and consists of quite stiff marls and calcareous marls, with expected thickness up to 250 mt.
GES unit (Late Miocene, evaporitic origin) consists of bituminous clays, sulfiferous limestones
and whitish nodular chalk banks. Also this formation outcrops along the coastline and has a
maximum thickness of 40-50m.
The *Colombacci* formation (FCO, late Miocene age) is mainly composed of clays and marly-
silty clays. The maximum thicknesses are greater than 100 m.
FAA formation (early Pliocene-early Pleistocene) widely outcrops in Ancona area (thickness
up to 300 m) and it is a pelitic succession that in its upper part consists of massive gray-blue
stratified marly clays with rare sand lenses. It is worth noting that this unit has strong lateral
and vertical variations.
The quaternary deposits in Ancona, according to the 282 sheet of the 1 : 50.000 Geological
Map of Italy (Lettieri, 2009), have been merged into the *Musone River* syntheme: the eluvial-
colluvial deposits (MUS$_{b2}$) cover sometimes large sectors of the hillsides, the surfaces of the
terraces, and fill the bottom of most of the valleys. Thickness can be up to 10-15m and they
consist of fine sediments (sands, clays and silts).
Quaternary slope instabilities (Agostini et al., 2014) affect areas at east and west of Ancona,
characterized by Plio-Pleistocene clay soils (e.g., Centamore et al., 1982; Cancelli et al., 2005;
Fiorillo 2003). The landslide deposits, whenever it was possible to represent them on a 1:25000



map, have been distinguished as unstable (MUSa1) or stable (MUSa1q). The Ancona landslide,
at west of Ancona, represents one of these instabilities.
The alluvial deposits (MUSbn) comprise the terraces and consist of heterometric silt-gravel
units. They are spread over the city of Ancona and their thickness is variable from point to
point but of the order of 15-50 m. In the more urbanized areas they can be completely covered
by anthropic sediments, 2m thick, consisting of coarse calcareous pebbles mixed to the old
natural soil.

### 2.2 EMERSITO INGV intervention

EMERSITO[6] is the INGV task force devoted to site effect and microzonation studies during
significant seismic crises in Italy. As for other INGV task forces[2], EMERSITO[6] is activated for
earthquakes exceeding magnitude 5.0 or whenever the observed damage is likely due to local
amplification effects. Since its official constitution in 2015, the group consists of a variable
number of people, to date about 50 INGV employees on a voluntary basis, among researchers,
technicians and technical collaborators, and involves various INGV departments and offices
spread in the italian territory. An operational protocol regulates the operation of the group,
organised by two national coordinators that lead a management team that includes a contact
person for each INGV office. EMERSITO[6] worked in the 2016-2017 Central Italy seismic
sequence (Cara et al., 2019; Priolo et al. 2020; Milana et al., 2020) and the 2017 Ischia
emergency (Nardone et al., 2023), but the group participated, in an unofficial form, also to
previous Italian emergencies (San Giuliano di Puglia 2002, Palermo 2002, L'Aquila 2009,
Emilia-Romagna 2012), increasing its experience in this research field.
From the beginning of the emergency, EMERSITO[6] started its activities by organizing itself in
specific working groups mainly to collect a variety of information regarding the epicentral area:
geology, damage surveys, previous studies on site effects and microzonation, seismic data by
nearby stations of the National Seismic Network run by INGV (Rete Sismica Nazionale-RSN;
INGV Seismological Data Centre, 2006) and the Italian Strong Motion Network run by the
Civil Protection (Rete Accelerometrica Nazionale-RAN, PCM-CPD, 1972). This information
has been uploaded in an online Web-GIS project (Fig. 5), shared and updatable in real time by
all the users located in different offices of INGV. This procedure was useful for sharing the
knowledge of the area and the ideas on the intervention through live and virtual meetings,
which guided the preliminary field inspections and the deployment of the seismic temporary
network.
The initial planning was carried out remotely considering the available Level 1 Seismic
Microzonation study, that incorporates noise measurements, downholes and boreholes with
stratigraphy (https://qmap-protciv.regione.marche.it/cs/) and the preliminary evidence of
earthquake-induced damage coming from the other INGV Task Forces (SISMIKO[3],
EMERGEO[4] and QUEST[5]). QUEST[5] in particular has provided first indications about the most
damaged areas in terms of affected buildings (Tertulliani et al., 2022): they reported a
macroseismic intensity of V EMS-98 for Ancona and individuated state of damage up to degree
3 in some buildings in downtown and damage 1-2 degree in a suburban neighbourhood for
some recent reinforced concrete buildings (vulnerability class C and D). Afterwards, the Fire
Brigade performed a detailed survey for all buildings, distinguishing the levels of damage in
the city (Fig. 2).
Ad hoc site inspections were carried out in collaboration with the INGV Ancona[7] office, which
has become a logistic support for all the task forces. It was then possible to contact several
institutions, i.e. the Marche Region (Albarello et al., 2022), the Regional Civil Protection, the



Municipality of Ancona and the Navy Headquarter in Ancona. They were really collaborative,
giving us suitable places for the station deployments, helping in finding further investigations
and technical reports in the vicinity of the sites. The final choice of the sites was also made on
the basis of fast single-station ambient noise measurements, in order to have a first-order
evaluation of possible resonance effects.
As aforementioned, the city suffered a low level of damage, then it did not have any major
impact on its usual activities. For this reason, installations inside buildings have been preferred
to guarantee continuous power supply and security of the seismic stations. We then identified
ground floors, basements or courtyards of private and public buildings, such as schools,
universities, sports centers, the Palace of the Regione Marche and religious structures.
Although EMERSITO[6] intervention was not focused on the landslide hazard, we decided to
install one station (CMA10) in the western part of Ancona, where the deep landslide moved in
240 1982.
After this preliminary phase, the final configuration of the temporary EMERSITO[6] network
covered the urban area of Ancona municipality and consisted of 11 six-channels digitizers,
coupled to velocimetric (Lennartz 3D-5 sec) and accelerometric (Kinemetrics Episensor)
sensors. Fig. 4 illustrates the position of the seismic stations in relation with the outcropping
geology, while Table 2 shows their location, coordinates, date of installation and data
transmission mode. The EMERSITO[6] temporary seismic network was registered in the
Federation of Digital Seismograph Networks (FDSN[9]) with the network code 6N[10]. At the same
time, station codes have been registered with the International Seismological Center (ISC[11]).
Most of the stations are installed close to the most damaged areas (compare with Fig. 2),
CMA06 is in the new industrial area in the south, CMA10 in the 1982 landslide area, close to
the district of Posatora.
A difficult task was the identification of sites characterized by the presence of outcropping stiff
lithologies where to install a reference station. After several tests, we found a possible reference
site on the so-called Colombacci formation (FCO), i.e. clay-marls of Miocene age, at about 90
mt from IV.PCRO station, free from clear resonance effects on noise, and installed the
reference station CMA15 (Figs 4 and 6).
The topography at Ancona downtown is not flat (Fig. 6). The medium elevation is about 70mt
but there are some hills that reach about 180-250 m and quickly slope towards the Adriatic sea.
Stations CMA15 and IV.PCRO are on a hill 140-160 m high whereas station CMA12 was
placed on the top of a hill 100 m high that quickly slopes towards the Adriatic sea and where
there is also the lighthouse of Ancona (Fig. 6). To avoid possible soil-interaction with the
lighthouse, the station was placed at about 30mt from it, inside a building of the Navy facilities.
**Table 2.** List of the sites of the 6N seismic network. The dismissing date of the stations was 24th of February
268 2023.

| Name | Location | Lat | Lon | Installation date | Acquisition mode |
|---|---|---|---|---|---|
| CMA05 | Piaget School | 43.618437 | 13.52708 | 2022-11-15 10:40 | Real Time |
| CMA06 | Paolinelli Sports Center, in the hamlet of Baraccola | 43.553738 | 13.511387 | 2022-11-15 11:32 | Real Time |
| CMA07 | Salesian Oratory | 43.605702 | 13.503745 | 2022-11-13 18:03 | Real Time |
| CMA08 | Economics University | 43.620228 | 13.516387 | 2022-11-14 15:12 | Real Time |





| CMA09 | Church of Saints Cosma and Damiano | 43.618237 | 13.515918 | 2022-11-13 11:12 | Real Time |
|---|---|---|---|---|---|
| CMA10 | Via della Grotta (landslide) | 43.603008 | 13.480115 | 2022-11-14 11:18 | Real Time |
| CMA11 | Navy | 43.598542 | 13.506017 | 2022-11-14 16:05 | Stand Alone |
| CMA12 | Cardeto park (lighthouse) | 43.622585 | 13.51589 | 2022-11-15 10:40 | Stand Alone |
| CMA13 | Via Barilatti | 43.593848 | 13.502273 | 2022-11-15 13:33 | Stand Alone |
| CMA14 | Raffaello Palace | 43.609948 | 13.509390 | 2022-11-15 16:07 | Stand Alone |
| CMA15 | Palascherma | 43.608372 | 13.531515 | 2022-11-15 16:08 | Stand Alone |

Figure 7 shows the 1D stratigraphic models under the installation sites, based on the available boreholes close to the stations and to our interpretation about the geological evolution of the area. The information used for the construction of these 1D stratigraphic models were located at a distance between 5 and 250 meters from the stations, determining different levels of reliability and uncertainty in the models, especially for the non-outcropping layers, considering the lateral variability and the different thickness and lithologies encountered.

The models reach a depth of 100 meters and are characterized by a variable thickness of altered/fractured layers. In particular, CMA06-CMA07-CMA11 stations, installed in flat valley areas, are composed of fine alluvial unconsolidated deposits (MUSb2) above the clayey formation of Argille Azzurre (FAA). CMA05-CMA08-CMA09-CMA13-CMA14-CMA15 stations are installed in quite flat areas and their stratigraphy featured by fine and more heterometric colluvial unconsolidated deposits (MUSb2, MUSbn) above the clayey (Argille Azzurre FAA) or marly (Schlier, SCH) or clayey/marly (Argille a Colombacci, FCO) geological formations. CMA10 is installed on the 1982 landslide sediments (MUSa1) whereas CMA12 is set on SCH formation in a topographic relief.

**3. Seismic data collection of the 6N network**

**3.1 Data availability**

The installation of the seismic stations was completed in three days and the 6N network was fully operative for 3 months, from November 13th, 2022, until February 24th, 2023.

The six stations in real-time acquisition mode (Table 2) transmitted data as well as their state of health (SOH), such as input voltage and quality of GPS signal received, to the EMERSITO[6] servers. Data availability and SOH were frequently checked with dedicated software tools. During the acquisition period, several maintenance interventions were carried out to download data from stand-alone stations and to verify their correct operation.

Raw data were converted into the standard binary *miniSEED* format, and organized in a structured seismic archive (following the SeisComP data structure). Then, data quality and completeness were checked, and all the relevant information was used for creating the metadata volumes with the perspective to upload them in the INGV node of the European Integrated Data Archive portal (EIDA[13]; Danecek et al., 2021).

All continuous data have been transferred to EIDA[13] and are currently available to everyone interested in. The dataset acquired by the EMERSITO[6] temporary network 6N[10] and described in this manuscript can be accessed under 10.13127/sd/qctgd6c-3a (EMERSITO Working Group, 2024), according to a set of rules defined by the INGV data management office (Open Data Portal-ODP[12]) and EMERSITO[6].

Figure 8 shows availability of recordings for each station of the 6N network as a function of time. The gaps in the records of some stations were caused by some malfunctions, in general





due to power failures; however, data completeness turned out to be quite satisfactory for all the
stations, being on average about 97%.
**3.2 Data quality**
In order to characterize the seismic background noise at the seismic stations of the temporary
EMERSITO[6] 6N[10] network, we computed the Power Spectral Density (PSD) using the three-
component continuous signals.
PSD and Probability Density Functions (PDF) were obtained from the waveform data and the
corresponding response files using the PPSD[14] class of ObsPy[15], a Python toolbox for
Seismology (Beyreuther et al., 2010), in which the computation of PSD and PDF is based on
the algorithm proposed by McNamara and Buland (2004). For each seismic channel, the
software computes the PDF from the distribution of the PSD values at each spectral interval,
providing the probability of occurrence of a given seismic signal level in a fixed frequency
interval.
We used the 90th percentile curves to get a robust estimate of the noise level and to compare it
between different stations, as shown in Figure 9 for the three components of motion. They are
often above the reference curves (new high and new low noise models, NHNM and NLNM
respectively) as computed by Peterson (1993). This was expected because the stations are
located in a highly urbanized area. The high noise level occurs mainly at frequencies above 1
Hz during day times, and there is a strong reduction of the noise level during night times (about
10-15 dB) and also during day times on Christmas holidays (by about 5 dB) (Fig. S1a in
Supplementary material).
The inspection of spectral and time amplitude levels allowed us to evaluate the suitability of
the installation sites and find critical situations. In particular, the CMA10 station was initially
installed inside a shelter that hosts electronic devices for monitoring movements of the active
landslide. This situation negatively affected the data quality of this station (Fig. S1b in
Supplementary material) with evident disturbances on the recordings. Consequently, the station
was moved outside the structure, about 2 meters away from the previous position, obtaining an
improvement in the data quality, with more stable and lower amplitude spectra (although some
artefacts are still present at about 20 sec).
**3.3 Recorded earthquakes**
During the operating time of network 6N[10] there were 258 aftershocks of the Marchigiana-
Pesarese seismic sequence with $2.0 \leq M \leq 2.9$, 28 with $3.0 \leq M \leq 4.0$ and 1 with $M = 4.2$, that
was the strongest one after the mainshock (Fig. 10a). Eight $M \geq 3.0$ events are related to other
local seismic sources in Italy located at a maximum distance of 100km from Ancona (Fig. 10b).
Of course not all the local events have been recorded by the stations of network 6N or, although
recorded, not all of them have a good quality.
Seven $M \geq 4.0$ events have an epicentral distance ranging from 100 to 500 km (Fig. 10c) and
the network was also able to record the strong Turkish earthquake that occurred the 6th of
February 2022 (Mwpd 7.9) at a distance of about 2200 km from Ancona (Fig. 10d).
Figure 11 shows an example of the $M_W$ 3.9 aftershock of December 8th at 07:08 UTC recorded
by some 6N[10] stations. The seismograms and the spectrograms highlight clear differences in
the site response: CMA12 and CMA15 sites, located on stiff units (FCO and SCH formations,
respectively), are characterized by short durations and small amplitudes, whereas stations
installed on poor sediments over stiffer materials (CMA10, CMA13 and CMA14) show longer
durations and higher amplitudes. The spectrograms also point out frequency variations.





Some differences can be also observed for low-frequency events, such as the teleseismic Mwpd
7.9 Turkish earthquake(Fig. 12).

**4. Preliminary analyses**
The recordings of ambient vibrations and earthquakes collected by the 6N[10] network allowed
us to perform some preliminary analyses for characterising the recording sites. Moreover, the
joint use of data of the temporary networks installed during the emergency, as the 6N one, and
of the permanent networks, in principle increase the chance to improve the estimates of the
earthquakes' parameters (i.e. their localization and focal mechanism).
We first present the different techniques used for the analyses and some illustrative results. The
overall results for each station of the network are presented as synthetic sheets collected in the
supplementary material.

**4.1 Localization and Focal mechanism improvements**
The availability of the local events recorded by network 6N[10], as well of other networks,
increase the chance to get better localization and to constrain the calculations of the focal
mechanisms, especially for the earthquakes where the first polarities can be depicted.
As an example, we used data of two events (see Table 3) recorded simultaneously by 3
networks: 6N[10], Y1 (managed by SISMIKO INGV emergency task force; D'Alema et al., 2022,
Moretti et al., 2023) and IV (RSN; INGV Seismological Data Centre, 2006). For event
#33466171 using only data from IV and Y1 it was not possible to calculate the focal
mechanism. Therefore we added the 6N data; first, using the phase picks from the seismograms,
we relocated the event by using a multi-parameter procedure (Ciaccio et al., 2021) that explores
the hypocenter solutions space by changing the *a-priori* key conditions that strongly influence
the solution convergence in the linearized approach. Then, we computed the double-couple
fault plane solutions from P-wave first motion data (FPFIT program, Reasenberg and
Oppenheimer, 1985). Finally, because our data allowed a significant increase of the sampling
of the focal sphere, the procedure successfully calculated the focal mechanism of the event
(Fig. 13). This focal mechanism shows a transpressive solution, is of good quality in terms of
uncertainties on strike, dip, rake (quality code QP= A) and station distribution ratio (STDR
<0.5), being this last quantity sensitive to the distribution of the data on the focal sphere
(Reasenberg and Oppenheimer, 1985).
The same procedure was followed for the event #33589291 (Table 3). In this case, the focal
solution was already available, but adding 6N data improved the STRD quantity (from 0.6 to
0.55) giving greater robustness to the solution.

**Table 3.** Location and focal mechanism parameters of the two analyzed seismic events. EventID: numerical
unique identifier of the INGV earthquakes database (http://terremoti.ingv.it).

| EventID | Date | Magnitude | Latitude | Longitude | Depth (km) | Strike | Dip | Rake |
|---------|------|-----------|----------|-----------|------------|--------|-----|------|
| 33466171 | 2022-11-23T01:59:26 | $M_L$ 3.6 | 43.9337 | 13.2537 | 15.75 | 100 | 50 | 30 |
| 33589291 | 2022-12-08T05:30:04 | $M_W$ 3.6 | 43.8975 | 13.2653 | 15.14 | 110 | 40 | 30 |

**4.2 Data analysis methods**



### 4.2.1 Horizontal-to-Vertical spectral ratio on noise (HVNSR) and earthquakes (HVSR)

The Horizontal-to-Vertical spectral ratio on noise (HVNSR) and earthquakes (HVSR) data play an important role in seismic microzonation and site effects studies (Hailemikael et al., 2020). Indeed they are widely used and can provide information on the resonance frequencies of the site, which is related to the thicknesses of the layers and their average shear wave velocity.

The HVNSR analysis (Nakamura, 1989), although not able to define the transfer function of the site, can provide useful indications on the possible resonance frequencies and on the susceptibility of a site towards possible amplification phenomena. To estimate the HVNSR at the Ancona network, we used the HVNEA software on the continuous recordings (Vassallo et al., 2023) which takes advantage of the Geopsy software (Wathelet et al., 2020). The computation results in hourly HVNSR curves as average on 120s windows and repeated over the entire duration of the acquisition (about 3 months). At the end, we produced 1.600 to 2.200 hourly HVNSR curves for each station.

The HVSR analysis is conceptually similar to HVNSR, but is performed on earthquakes rather than on noise. Similarly to HVNSR, HVSR was performed with the software HVNEA, described in Vassallo et al. (2023). For each event, HVSR is calculated on a 6-second window from the theoretical S-wave arrival time. The averages were obtained by using a subset of events from the INGV earthquake bulletin[16], using a circular search of magnitude M>=3 events at a maximum distance of 50 km from Ancona city (Table 4). With these criteria, the considered earthquakes had a signal-to-noise ratio (SNR) >=3 in the frequency range 0.5-15.0 Hz. The number of selected events ranges from 17 to 29, then the results are indicative.

**Table 4.** List of the earthquakes used for HVSR and SSR analysis

| #EventID | Time | Latitude (degrees) | Longitude (degrees) | Depth (Km) | Author | MagType | Magnitude | EventLocationName |
|---|---|---|---|---|---|---|---|---|
| 33378441 | 2022-11-14T23:10:54.960000 | 43.9368 | 13.3483 | 5.2 | BULLETIN-INGV | $M_L$ | 3.5 | Costa Marchigiana Anconetana (Ancona) |
| 33389921 | 2022-11-16T08:57:08.040000 | 43.934 | 13.337 | 4.4 | SURVEY-INGV | $M_L$ | 3.2 | Costa Marchigiana Anconetana (Ancona) |
| 33418361 | 2022-11-19T03:56:03.320000 | 43.9767 | 13.3195 | 10.8 | SURVEY-INGV | $M_L$ | 3.0 | Costa Marchigiana Pesarese (Pesaro-Urbino) |
| 33431491 | 2022-11-20T05:20:30.250000 | 43.9027 | 13.2642 | 10.3 | SURVEY-INGV | $M_W$ | 4.2 | Costa Marchigiana Pesarese (Pesaro-Urbino) |
| 33431631 | 2022-11-20T05:23:19.770000 | 43.9677 | 13.3185 | 8.7 | SURVEY-INGV | $M_L$ | 3.2 | Costa Marchigiana Pesarese (Pesaro-Urbino) |
| 33434911 | 2022-11-20T09:59:46.700000 | 43.9083 | 13.3353 | 9.2 | SURVEY-INGV | $M_L$ | 3.3 | Costa Marchigiana Anconetana (Ancona) |
| 33435461 | 2022-11-20T10:38:54.300000 | 43.9625 | 13.2825 | 7.9 | SURVEY-INGV | $M_L$ | 3.3 | Costa Marchigiana Pesarese (Pesaro-Urbino) |
| 33466171 | 2022-11-23T01:59:26.800000 | 43.91 | 13.2288 | 10.2 | BULLETIN-INGV | $M_L$ | 3.6 | Costa Marchigiana Pesarese (Pesaro-Urbino) |





| | | | | | | | | |
|---|---|---|---|---|---|---|---|---|
| 33477031 | 2022-11-24T17:26:40.160000 | 43.925 | 13.2753 | 9.1 | SURVEY-INGV | $M_L$ | 3.2 | Costa Marchigiana Pesarese (Pesaro-Urbino) |
| 33477901 | 2022-11-24T22:11:30.200000 | 43.904 | 13.2937 | 9.5 | SURVEY-INGV | $M_L$ | 3.2 | Costa Marchigiana Pesarese (Pesaro-Urbino) |
| 33533041 | 2022-12-01T00:03:02.130000 | 43.8888 | 13.3305 | 9.7 | SURVEY-INGV | $M_L$ | 3.4 | Costa Marchigiana Anconetana (Ancona) |
| 33534141 | 2022-12-01T04:42:07.310000 | 43.8875 | 13.339 | 8.8 | SURVEY-INGV | $M_L$ | 3.2 | Costa Marchigiana Anconetana (Ancona) |
| 33584401 | 2022-12-07T11:06:10.980000 | 43.9202 | 13.3133 | 10.0 | SURVEY-INGV | $M_L$ | 3.0 | Costa Marchigiana Pesarese (Pesaro-Urbino) |
| 33589291 | 2022-12-08T05:30:05.540000 | 43.913 | 13.297 | 9.1 | BULLETIN-INGV | $M_W$ | 3.6 | Costa Marchigiana Pesarese (Pesaro-Urbino) |
| 33590351 | 2022-12-08T06:55:41.970000 | 43.954 | 13.3127 | 9.1 | SURVEY-INGV | $M_L$ | 3.0 | Costa Marchigiana Pesarese (Pesaro-Urbino) |
| 33590571 | 2022-12-08T07:08:18.650000 | 43.914 | 13.2888 | 8.4 | BULLETIN-INGV | $M_W$ | 3.9 | Costa Marchigiana Pesarese (Pesaro-Urbino) |
| 33591681 | 2022-12-08T08:06:50.860000 | 43.9312 | 13.3175 | 8.9 | SURVEY-INGV | $M_L$ | 3.3 | Costa Marchigiana Pesarese (Pesaro-Urbino) |
| 33645871 | 2022-12-14T08:34:05.690000 | 44.0173 | 13.2392 | 9.1 | SURVEY-INGV | $M_L$ | 3.0 | Costa Marchigiana Pesarese (Pesaro-Urbino) |
| 33683471 | 2022-12-19T07:37:13.480000 | 43.8762 | 13.3748 | 8.8 | SURVEY-INGV | $M_L$ | 3.3 | Costa Marchigiana Anconetana (Ancona) |
| 33771681 | 2022-12-31T00:37:35.720000 | 43.9827 | 13.3077 | 8.8 | SURVEY-INGV | $M_L$ | 3.1 | Costa Marchigiana Pesarese (Pesaro-Urbino) |
| 33804101 | 2023-01-04T15:55:18.660000 | 43.939 | 13.275 | 9.5 | BULLETIN-INGV | $M_L$ | 3.5 | Costa Marchigiana Pesarese (Pesaro-Urbino) |
| 33804361 | 2023-01-04T16:01:18.420000 | 43.9262 | 13.2773 | 8.7 | SURVEY-INGV | $M_L$ | 3.3 | Costa Marchigiana Pesarese (Pesaro-Urbino) |
| 33870151 | 2023-01-12T07:06:14.500000 | 43.9117 | 13.2668 | 9.6 | BULLETIN-INGV | $M_L$ | 3.6 | Costa Marchigiana Pesarese (Pesaro-Urbino) |
| 33959201 | 2023-01-21T18:52:37.040000 | 43.9348 | 13.3682 | 7.7 | SURVEY-INGV | $M_L$ | 3.2 | Costa Marchigiana Anconetana (Ancona) |
| 33977501 | 2023-01-25T14:30:20.590000 | 43.9682 | 13.3052 | 7.9 | SURVEY-INGV | $M_L$ | 3.0 | Costa Marchigiana Pesarese (Pesaro-Urbino) |
| 34020401 | 2023-02-02T04:18:22.520000 | 43.9823 | 13.3227 | 7.0 | SURVEY-INGV | $M_L$ | 3.2 | Costa Marchigiana Pesarese (Pesaro-Urbino) |





| 34024531 | 2023-02-02T14:49:37.610000 | 43.9583 | 13.2907 | 7.2 | SURVEY-INGV | $M_L$ | 3.1 | Costa Marchigiana Pesarese (Pesaro-Urbino) |
| 34161341 | 2023-02-21T00:07:20.490000 | 43.2798 | 13.3392 | 7.4 | BULLETIN-INGV | $M_W$ | 3.6 | 1 km NW Pollenza (MC) |


### 4.2.2 *Directional amplification in frequency and time domain*
Directional amplification effects imply that there is a preferential direction of amplification of
the horizontal Fourier spectra, reported as a strike from the geographic north, as firstly proposed
by Bonamassa and Vidale (1991). In the time domain, they correspond to linearly polarized
ground motion, with mean polarization along the direction of maximum amplification.
In this work, directional amplification effects are preliminarily investigated in the frequency
domain through the calculation of rotated horizontal-to-vertical spectral ratios both on noise
(HVNSR) and earthquakes (HVSR), and in the time domain by using the covariance matrix
analysis (Kanasewich, 1980; Jurkevics 1988).
The use of rotated spectral ratios was first introduced by Spudich et al. (1996) and subsequently
exploited by several authors to detect the horizontal polarization of ground motion on
topography and in fault zones (e.g., Rigano et al., 2008; Di Giulio et al., 2009; Pischiutta et al.,
2012) or on sedimentary basins (Theodoulidis et al., 2018).
For the computation on noise, we used the Geopsy software (Whatelet et al., 2020) applying
an anti-trigger algorithm to select the most stationary part of the signals, as well as a cosine
taper and a Konno-Ohmachi smoothing filter with coefficient b = 40 (Konno and Ohmachi,
1998). We calculated HVNSR after rotating the NS and EW components by steps of 10°, from
0° to 180°.
For earthquakes we considered the same list in Table 4 used for HVSR analysis. We first cut a
portion of each event, a 6-seconds long window, including the S and early coda waves. Then,
we computed the direction of maximum amplification as the azimuth at which the HVSR peak
reaches the maximum value. Conventionally, the directional amplification effect is considered
significant if the ratio between the maximum and minimum amplitude levels at the frequency
peak exceeds 1.5 (Pischiutta et al., 2018). The complete values retrieved by the rotated HVNSR
and HVSR are given in the Supplementary material (Tables S1 and S2, corresponding to results
from earthquake and ambient noise recordings, respectively).
The covariance matrix method in the time domain (Jurkevics, 1988) is an alternative method
to estimate the ground motion polarization both on noise and earthquakes, in particular when
directional peaks have been observed with the rotated HVNSR or HVSR. The method results
in the estimation of the polarization ellipsoid. In order to give a quantitative evaluation on how
much elongated the polarization ellipsoids is, we apply the hierarchical criterion proposed by
Pischiutta et al. (2012), which results are given in the supplementary material (Tables S1 and
S2, corresponding to results from earthquake signals and ambient noise, respectively).
### 4.2.3 Horizontal-to-Horizontal spectral ratio (SSR)
The Horizontal-to-Horizontal spectral ratios (SSR) technique is based on the assumption that
the ratio between horizontal Fourier spectra from earthquakes recorded at a given site and at a
bedrock site represent a good estimate of the transfer function of the site. The implicit
assumption is that the contribution of the source and the crustal propagation is the same for the
two sites, and that the spectrum of the rock site (i.e. the reference station) is free from
amplification effects (Borcherdt, 1970; Cara et al., 2011). For these reasons, this technique is



believed to give the seismic response of a given site, not only limited to the resonance effects
as for HVNSR or HVSR.
For network 6N[10] we chose CMA15 station as the most suitable reference site, being installed
on an outcropping geological bedrock (FCO, Colombacci Formation). Moreover, its recordings
are characterized by short duration, small amplitudes and no resonance frequency peaks (see
Figures 11 and 14).
In order to automate the calculation, a script implemented in a Python environment and based
on the ObsPy[15] framework (Beyreuther et al., 2010) was used. The code allows to: (1) extract
the signal related to a seismic event over a time window of definable duration (6s in this case)
starting from the arrival of the S wave, which has been estimated using the technique proposed
by Akazawa (2004); (2) calculate the signal-to-noise ratio (SNR); (3) process the signals with
a Konno and Ohmachi (1998) filter and, finally, calculate the SSR ratios. The iterative
application was applied on the same list of HVSR analysis taking into account the simultaneous
presence of events on both the considered site and the reference site (Table 4).
**4.3 Summary results**
This subsection illustrates the results of the techniques described in the previous sections, by
using three selected stations as representative of the network: CMA08, CMA14 and CMA15.
The results for all the stations of the 6N network are given as synthetic sheets and collected in
the supplementary material (Figures from S3 to S13).
Figure 14 shows the HVNSR, HVSR and SSR results for the three considered stations. In the
following we summarize some preliminary conclusions:
a)  HVNSR amplitudes are relatively low (about 2 in average) and no clear resonance
peaks are observed.
b)  HVNSR and HVSR of station CMA15 are flat, as expected for a reference site.
c)  HVSR curves of CMA08 and CMA14 are slightly different from HVNSR ones: the
amplitudes are higher and also the frequency peaks depicted by the two techniques are
different. It should be considered that the number of earthquakes used for HVSR is not
very high, therefore the result is only indicative.
d)  SSR analysis shows very different outcomes than HVSR analysis. This behavior could
be due to the choice of the reference site (CMA15), and/or to possible 2- or 3-
dimensional site effects not accounted for by the HVSR technique.
The analysis of HVNSR carried out over the entire recording period was also important to
assess the temporal stability of the spectral peaks at each site (see Fig. S2 in Supplementary
material). There was no relevant variation of the peak frequencies whereas the peak amplitude
shows temporal variations up to 20%. These variations are mostly related to day-night spectral
levels reduction, especially in the vertical components and above 4 Hz.
Results of directional and polarization analyses, on both earthquake and noise, are shown in
Figure 15 for two stations, CMA08 and CMA14.
For station CMA08 the rotated HVNSR and HVSR highlights the presence of a directional
peak at about 3-4 Hz, and along N90°-110° azimuth (roughly, E-W direction). The pattern is
more complex at station CMA14 (Fig. 15, bottom panels), where earthquakes and noise give
slightly different outcomes. Earthquake recordings show two clear peaks in the HVSR analysis,
the former at 2.6 Hz, with maximum amplification roughly N-S and the latter at 4.4 Hz that is
not directional. Circular histograms of polarization azimuths obtained from filtered earthquake
signals in the frequency band 1-3 Hz, show a similar trend in N-S direction.



## 6. Data Availability
Data described in this manuscript can be accessed under 10.13127/sd/qctgd6c-3a (EMERSITO
Working Group, 2024).
## 7. Discussion and conclusions
The aims of this work were to illustrate the seismic dataset collected by the 6N temporary
network at Ancona, stored and available from the EIDA database, describe the intervention of
the EMERSITO working group and focus on the difficulties that can be encountered in urban
contexts during emergency activities, and finally to present the preliminary results that can be
achieved during a seismic sequence.
The overall results of HVSR and polarization analysis on both earthquakes and noise are
summarized in Figure 16.
As aforementioned, the HV on noise does not detect some frequency peaks, which are evident
only by earthquake data (CMA05, CMA06, CMA09, and CMA14), and, for some other peaks,
displays lower amplitude and/or no directionality (CMA05, CMA07, CMA09, CMA12,
CMA14). HVNSR and HVSR for station CMA10, which is set on the 1982 landslide, have a
shape with no clear resonance peak.
In terms of directional motion the results between noise and earthquakes are fully consistent
only at stations CMA08, CMA11, and CMA15.
Table 5 lists, for each 6N[10] station, the outcropping lithology, the number of peaks observed
on HVSRs and for each one, the peak frequency and amplitude values. When amplification is
found to be directional, the direction of maximum amplification and polarization is given as
well.
The lowest resonance frequency value from data analysis (Table 5), observed at the sites
CMA07, CMA11 and CMA15, is around 1.5 Hz (frequency range 1-2.5 Hz in Fig. 16) and
related to thick clay deposits (Fig. 7). The majority of sites show $f_0$ values in the range 2.5-5
Hz. Higher frequencies ($f_0 > 5$ Hz) are observed at two stations (CMA12 and CMA05) closest
to the sea in the northern direction, where the Schlier marly Formation is nearly outcropping
(Fig. 7).
**Table 5.** Synthesis of results of directional analysis (frequency and amplitude values of resonance peaks) obtained
from HVSR and HVNSR analysis.

| | Summary of HVSR and HVNSR analyses | | | | | | |
|---|---|---|---|---|---|---|---|
| **Station** | **Site conditions** | **N. peaks** | **#** | **Frequency peak (Hz)** | **Ampl.** | **Direction max ampl. (degrees)** | **Notes** |
| **CMA05** | **SCH - Schlier Fm.** | 2 | 1 | 5.2☐5.6 | 2.7☐4.1 | 30☐36 | HVSRs indicate no directionality |
| | Marly limestones and clays (Miocene) | | 2 | 9.7 | 2.8☐3.6 | 12☐20 | Peak evident only on HVSRs |





| | | | | | | |
|---|---|---|---|---|---|---|
| **CMA06** | **MUSbn - Musone Fm.** | 2 | 1 | 1.2⎯1.3 | 2.4⎯2.9 | none | |
| | Terrace deposits (Holocene) | | 2 | 3.5⎯3.7 | 2.7⎯4.7 | none | Peak evident only on HVSRs |
| **CMA07** | **MUSbn - Musone Fm.** | 1 | 1 | 1.6⎯2.2 | 2.1⎯3.5 | 30⎯60 | HVNSRs have lower amplitudes than HVSRs |
| | Terrace deposits (Holocene) | | | | | | |
| **CMA08** | **Musb2- Musone Fm.** | 1 | 1 | 2.8⎯3.9 | 2.3⎯3.5 | 80⎯110 | |
| | Eluvio-colluvial deposits (Holocene) | | | | | | |
| **CMA09** | **Musb2 Musone Fm.** | 2 | 1 | 1.7⎯2.4 | 2.1⎯3.3 | 170 | HVNSRs have lower amplitudes than HVSRs and no directionality |
| | Eluvio-colluvial deposits (Holocene) | | 2 | 3.5⎯3.7 | 2.7⎯4.7 | 80 | Peak evident only on HVSRs |
| **CMA10** | **Musa1 - Musone Fm.** | 3 | 1 | 2.6⎯2.7 | 2.1 | none | Peak evident only on HVSRs |
| | Active landslide deposits | | 2 | 4.1⎯4.4 | 2.3 | 0 | Peak evident only on HVSRs |
| | (Holocene) | | 3 | 5.3⎯7.5 | 2⎯3.2 | none | Broadband peak |
| **CMA11** | **MUSbn - Musone Fm.** | 1 | 1 | 1.4⎯1.5 | 2.1⎯3.3 | none | |
| | Terrace deposits (Holocene) | | | | | | |
| **CMA12** | **SCH - Schlier Fm.** | 1 | 1 | 8.8⎯9.6 | 2.5⎯3.7 | 100 | HVSRs indicate no directionality |
| | Marly limestones and clays (Miocene) | | | | | | |
| **CMA13** | **MUSbn - Musone Fm.** | 1 | 1 | 1.4⎯2.6 | 2⎯3.6 | 10 | HVNSRs have lower amplitudes than HVSRs and no directionality |
| | Terrace deposits (Holocene) | | | | | | |
| **CMA14** | **FAA - Argille Azzurre Fm.** | 2 | 1 | 2.2⎯2.6 | 2⎯2.7 | 140⎯170 | HVNSRs have lower amplitudes than HVSRs |
| | Marly and silty clays (Pleistocene) | | 2 | 4.4⎯4.5 | 2.5⎯3.2 | none | Peak evident only on HVSRs |
| **CMA15** | **FCO - Colombacci Fm.** | no peaks | | | | | |
| | Marly clays with conglomeratic levels (Miocene) | | | | | | |

However, it is important to say that for a complete geological-based interpretation, the earthquake database collected during the experiment needs to be fully analyzed, with a detailed search of M<3.0 events with SNR>=3, to have more robust statistics.

At the stage of the activities of EMERSITO during the seismic sequence, we can infer some points to be investigated in detail in future papers:

a) The HVNSR technique was a good method to test the functioning of the stations and the variability in an urban context, but it seems that for this case study, where the geological features do not show strong impedance contrast, is not very suitable for revealing resonance effects.

b) Also the HVSR technique, even if it has to be refined with a greater number of earthquakes, shows similar trends of HVNSR but with higher amplitudes and more evident peaks.

c) The SSRs are strongly different from HVNSR and HVSR. Also SSR has to be refined with a greater number of earthquakes, but the role of the reference station needs to be investigated. If the SSRs will result reliably, the next step will be to compare these amplification estimates with numerical simulations based on the available geological profiles for each site. Therefore, the use of 1D, 2D and maybe 3D simulations hopefully will explain the observed amplification pattern.

d) Although the role of landslide sediments in the amplification pattern is out of the aim of this work, we believe that specific and multidisciplinary studies based on extensive measurements in the unstable zones of the city are needed. It has to be taken into account that in unfavorable hydrological conditions, seismic waves of a possible moderate-to-strong earthquake could trigger the landslide movements.

e) All the stations (except CMA06 and CMA14 situated in external courtyards) are installed in the basement floors into buildings, then the interaction between soil and structures can have played a role in the observed results.



**Acknowledgements.** We thank the people of Ancona who hosted the instruments. In particular we thank the Navy and the Regione Marche for all the support, facilities and information that they have made available.
We gratefully acknowledge the Laboratory ESITO of INGV (https://www.ingv.it/monitoraggio-e-infrastrutture/laboratori/laboratorio-effetti-di-sito) for the technical support during the experiment.
We also thank the Italian Department of Civil Defence (DPC) for the economic support and a special thanks to the INGV group, in particular Massimo Fares, Diego Franceschi and Ivano Carluccio that helped us in the procedure to share the data of the network 6N in EIDA and Mario Locati for the INGV data policy task.

**Details on dataset access.**

**The best way for downloading data from EIDA is to use the Orfeus Data Center WebDC3 Web Interface**

**Repository**: http://www.orfeus-eu.org/webdc3/

Go to the "Explore Stations" tab, set Network Type as "All temporary nets" and Network Code as "6N*+ (2022) - Emersito Seismic Network in Ancona (Central Italy)". Select the HN (velocimetric data) or EH (accelerometric data) channels or both. Then press "Search". A list of the available seismic stations appears, it is possible to select all or only the desired stations.

Go to the "Submit request" tab and set the appropriate Time Selection Window. If you wish to download the complete records, set the time windows from 9-11-2022 to 28-02-2023 which include the whole recording period of the network 6N. Unfortunately the Orfeus Data Center limits the maximum size of data that can be downloaded for each request at about 1.5Gb. This means that it is possible to download up to 30 days of the HN (velocimetric) channels or up to 15 days of the EH (accelerometric) channels of one station at a time. Anyhow, to reduce the waiting times we suggest halving the request, e.g. 15 days for station and HN channels.

You can also choose to request for the miniseed data only, the metadata in XML format or the metadata in text format.

If everything is ok go to the "Download Data" tab, where you can follow the status of the FDSNWS requests. At the end click on the "SAVE" button to download the requested data.

**Author's contribution**: F. Cara, G. Di Giulio, M. Vassallo, G. Cultrera, G. Riccio, S. Lovati and F. Pacor are the coordinators of the Emersito task force. They designed and managed the experiment, so they contributed to the project administration and the conceptualization tasks. D. Famiani has been charged as scientific manager for the experiment, supervised by the coordinators. She wrote the initial draft of this manuscript that was revised and completed by the coordinators, in particular by F. Cara, G. Cultrera, G. Di Giulio and F. Pacor.

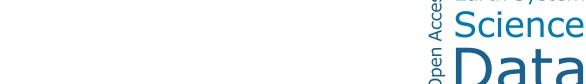



G. Di Giulio, M. Vassallo, D. Famiani, G. Brunelli, A. Bobbio, M. Pischiutta, S.
Hailemikael, A. Mercuri, G. Milana, L. Minarelli, A. Di Filippo, L. Nardone, S.
Marzorati, C. Ladina, D. Pantaleo, and C. Calamita contributed to the investigation,
finding the sites, deploying the seismic stations and maintaining them.
M. Vassallo, G. Riccio, A. Costanzo, A. Bobbio, M. Pischiutta, M. Massa, R. Puglia,
S. Hailemikael, A. Mercuri, G. Milana, M.G. Ciaccio, S. Pucillo, G. Sgattoni and C.
Ladina contributed to the formal analysis.
G. Riccio was in charge of data curation.
G. Brunelli contributed to the definition of 1D stratigraphy models under the
investigated sites.
R. Cogliano contributed to the maintenance of the web-gis whereas S. Pucillo, A.
Fodarella, G. Brunelli and D. Famiani helped in finding resources to add to the web-
gis.
G. Mele and C. Bottari helped the coordinators in the initial dissemination of the
experiment, useful also for the writing of this manuscript.
L. Falco G. and A. Memmolo contributed to the instrumental part, in particular in the
setting of the real-time stations.
M. Massa, G. Mele and C. Bottari, G. De Luca, G. Sgattoni and G. Tarabusi contributed
to the initial ideas about the experiment and also to the resources.
**Competing interests**: The authors declare that they have no conflict of interest.



**Footnotes**
[1] https://www.ingv.it/en/index.php
[2] https://www.ingv.it/en/monitoring-and-infrastructure/emergencies/emergency-groups
[3] https://sismiko.ingv.it/
[4] https://emergeo.ingv.it
[5] https://quest.ingv.it
[6] http://emersitoweb.rm.ingv.it/index.php/it/
[7] http://www.an.ingv.it/
[8] https://esm-db.eu/#/event/INT-20221109_0000046
[9] https://www.fdsn.org/
[10] https://fdsn.org/networks/detail/6N_2022/
[11] http://www.isc.ac.uk
[12] https://data.ingv.it/en/
[13] https://eida.ingv.it/en/
[14] https://docs.obspy.org/packages/autogen/obspy.signal.spectral_estimation.PPSD.html
[15] https://docs.obspy.org/
[16] http://terremoti.ingv.it/en/







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



**Figures**

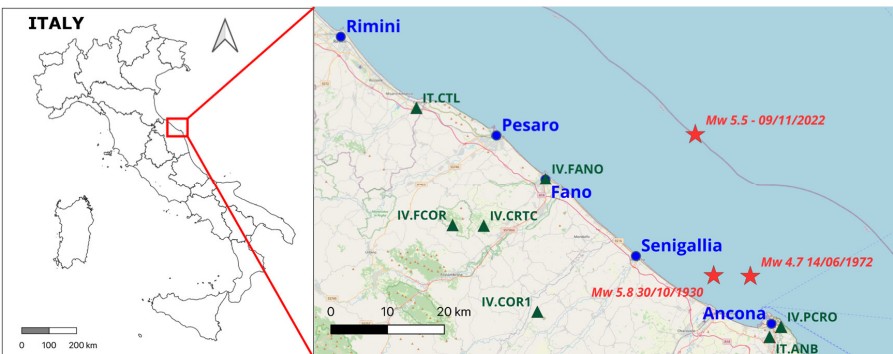

**Figure 1**. Left: Map of Italy, the red square indicates the Costa Marchigiana-Pesarese. Right: zoom of the study area
showings: a) the epicenter of the MW 5.5 of 09/11/2022 event, and the epicenters of the two strongest earthquakes
occurred in the previous century that affected Ancona significantly (red stars); b) the main cities in the Adriatic coast
(blue dots); c) the accelerometric stations (green triangles) of RAN and RSN seismic networks closest to the MW
5.5 event.
© OpenStreetMap contributors 2024. Distributed under the Open Data Commons Open Database License (ODbL)
v1.0.






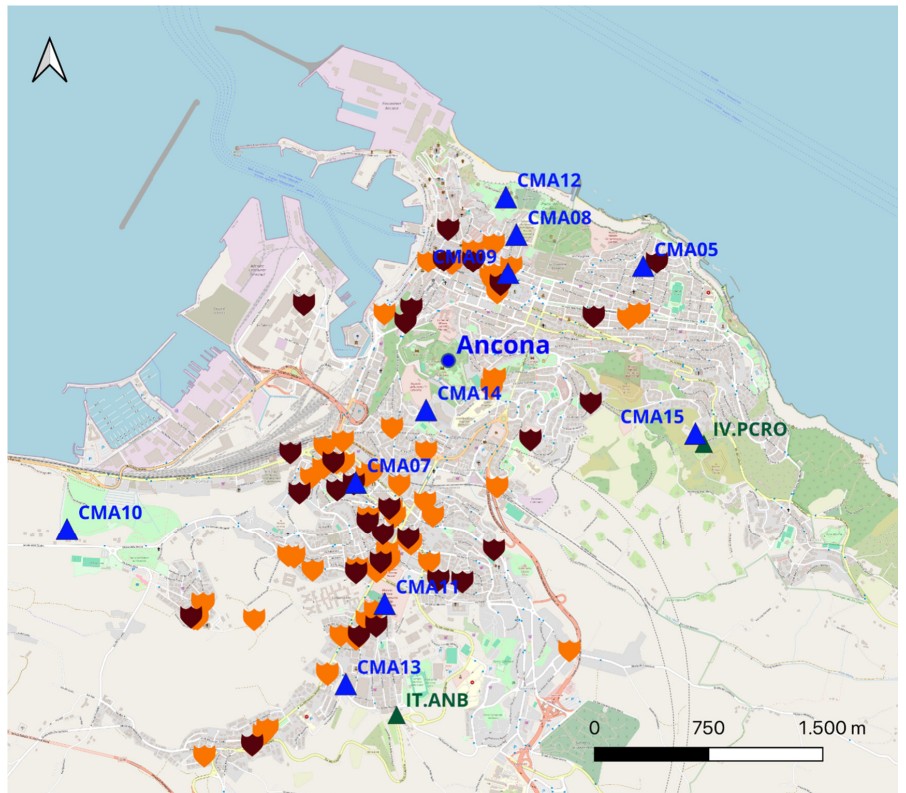

**Figure 2.** Map of Ancona municipality with the indication of damage reported by the Fire Brigades (from orange to dark-red symbols for increasing intensity, respectively). The blue triangles are most of the stations of the temporary network 6N installed by the EMERSITO working group. The green triangle are the two permanent stations installed at Ancona, IT.ANB and IV.PCRO, respectively.
© OpenStreetMap contributors 2024. Distributed under the Open Data Commons Open Database License (ODbL) v1.0.





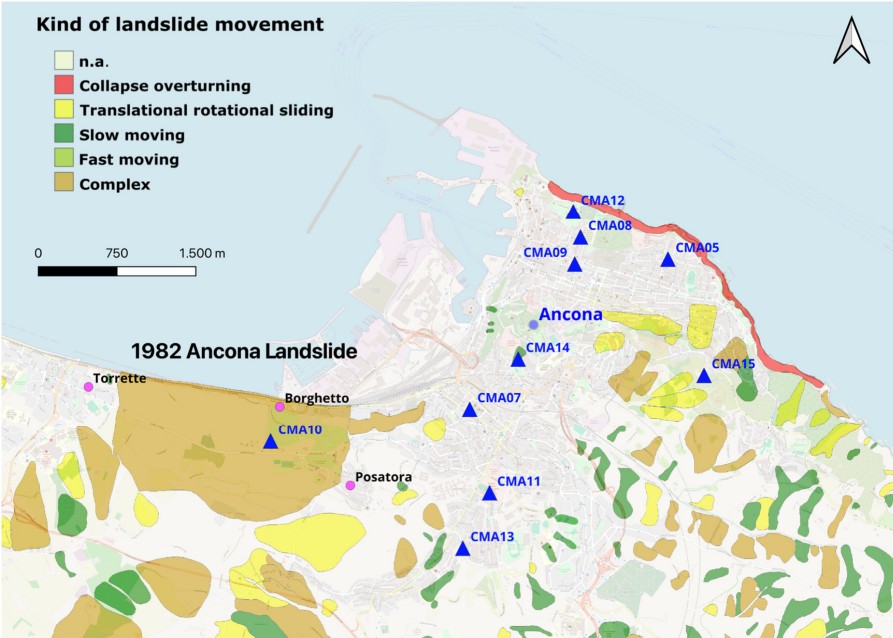

**Figure 3.** Map of Ancona municipality with landslide phenomena, as carried out by Italian Institute for Environmental Protection and Research (ISPRA) and the Italian Regions and Autonomous Provinces during the project IFFI (Inventory of Landslide Phenomena in Italy). In the map the huge area of the 1982 landslide is highlighted. The magenta dots represent the three districts of Ancona involved in the landslide movement. The blue triangles are most of the stations of the temporary network 6N installed by the EMERSITO working group. The green triangle are the two permanent stations installed at Ancona, IT.ANB and IV.PCRO, respectively. The complete IFFI database is available at the website:
https://www.isprambiente.gov.it/it/progetti/cartella-progetti-in-corso/suolo-e-territorio-1/iffi-inventario-dei-fenomeni-franosi-in-italia.
© OpenStreetMap contributors 2024. Distributed under the Open Data Commons Open Database License (ODbL) v1.0.

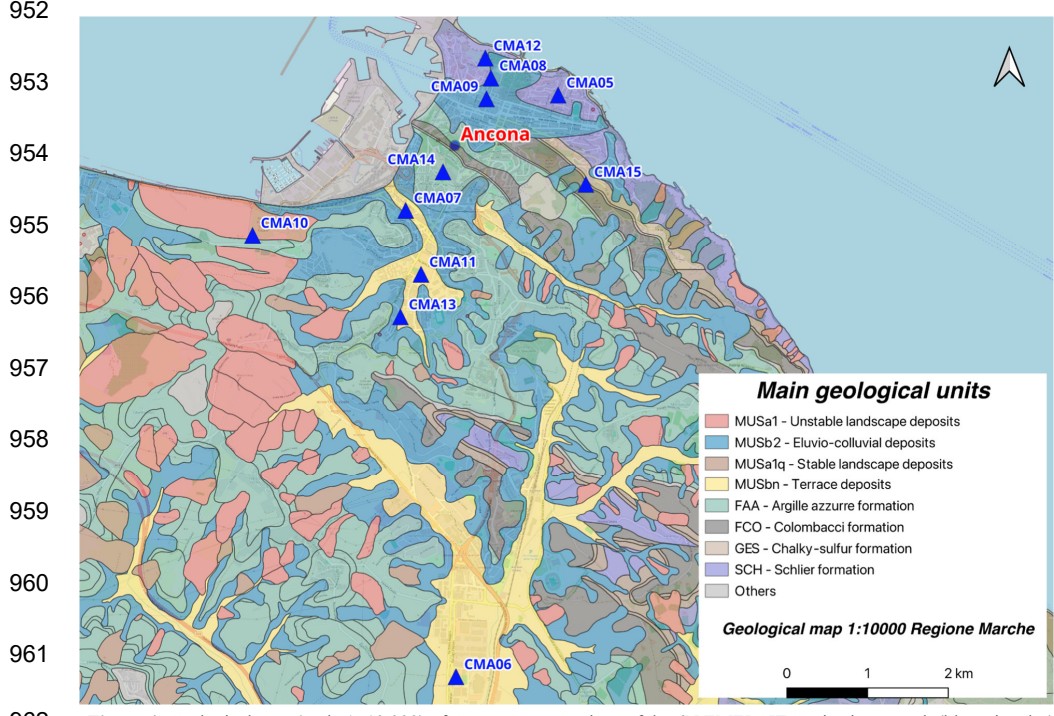

**Figure 4.** Geological map (scale 1: 10.000) of Ancona area. Stations of the 6N EMERSITO seismic network (blue triangles) are superimposed.
© OpenStreetMap contributors 2024. Distributed under the Open Data Commons Open Database License (ODbL) v1.0.



965

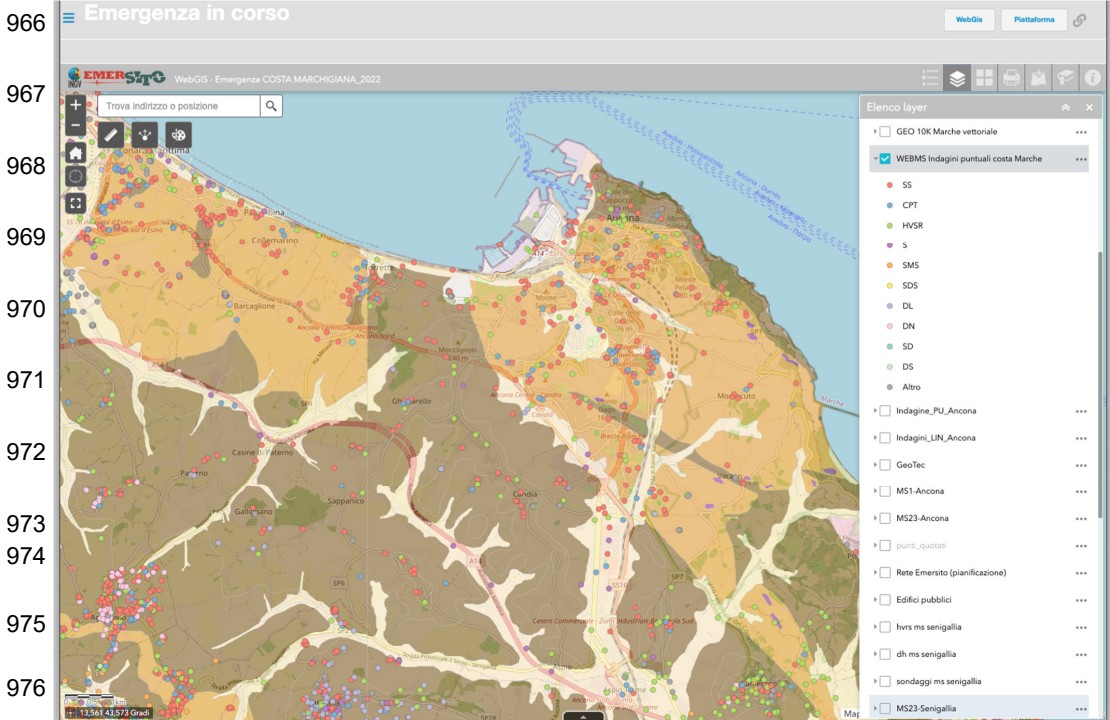

**Figure 5.** Example of layout used in the online Web-GIS project of EMERSITO, showing the Adriatic coast of Ancona, the lithological map and the available surveys used in microzonation studies (coloured dots).

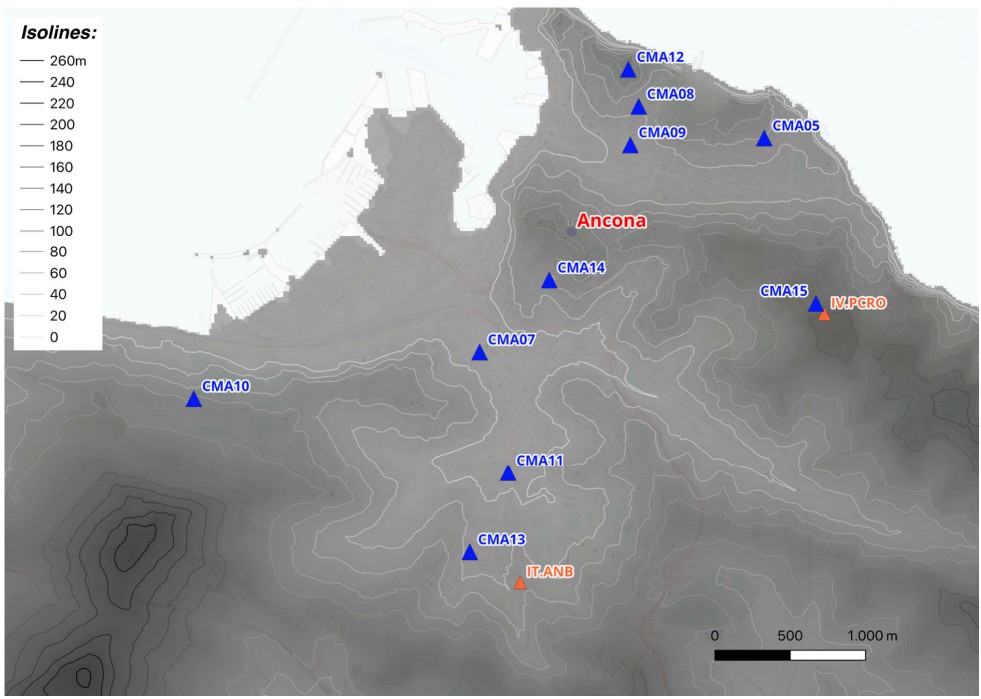

**Figure 6.** Topography map with isoline of the Ancona area. The blue triangles are most of the stations of the 6N EMERSITO Network, the orange triangles are the two permanent stations of RAN (IT.ANB) and RSN (IV.PCRO). Tarquini S., Isola I., Favalli M., Battistini A. (2007). © TINITALY, a digital elevation model of Italy with a 10 meters cell size (Version 1.0) [Data set]. Istituto Nazionale di Geofisica e Vulcanologia (INGV). https://doi.org/10.13127/tinitaly/1.0




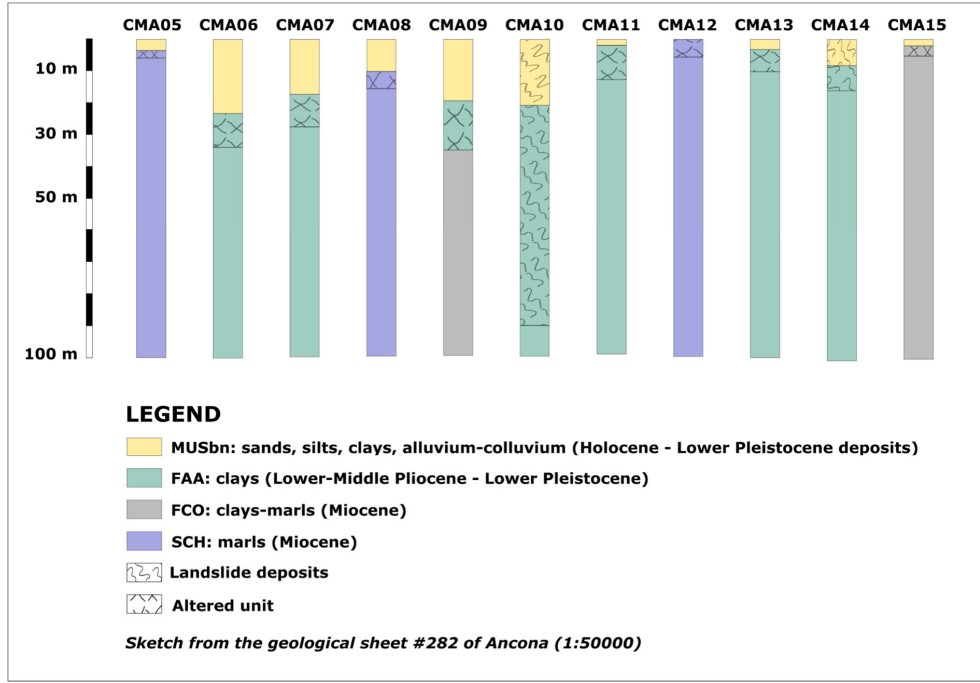

**Figure 7.** 1D stratigraphic models derived at the sites where 6N seismic stations are located.






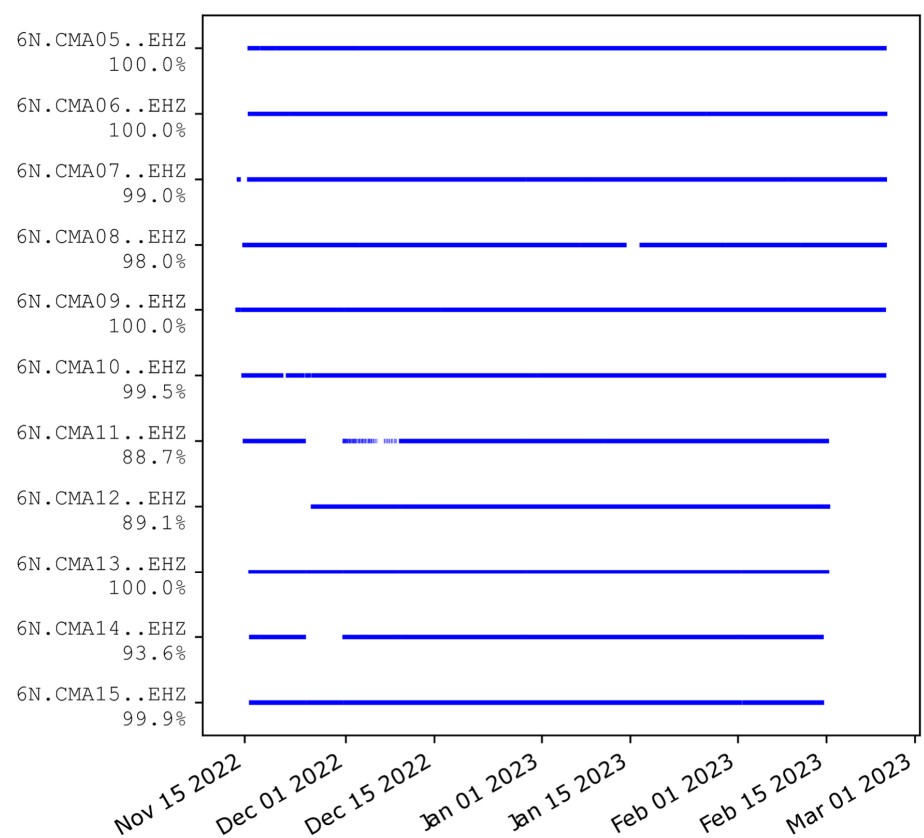

**Figure 8.** Data availability of the stations of the 6N network during the experiment period.

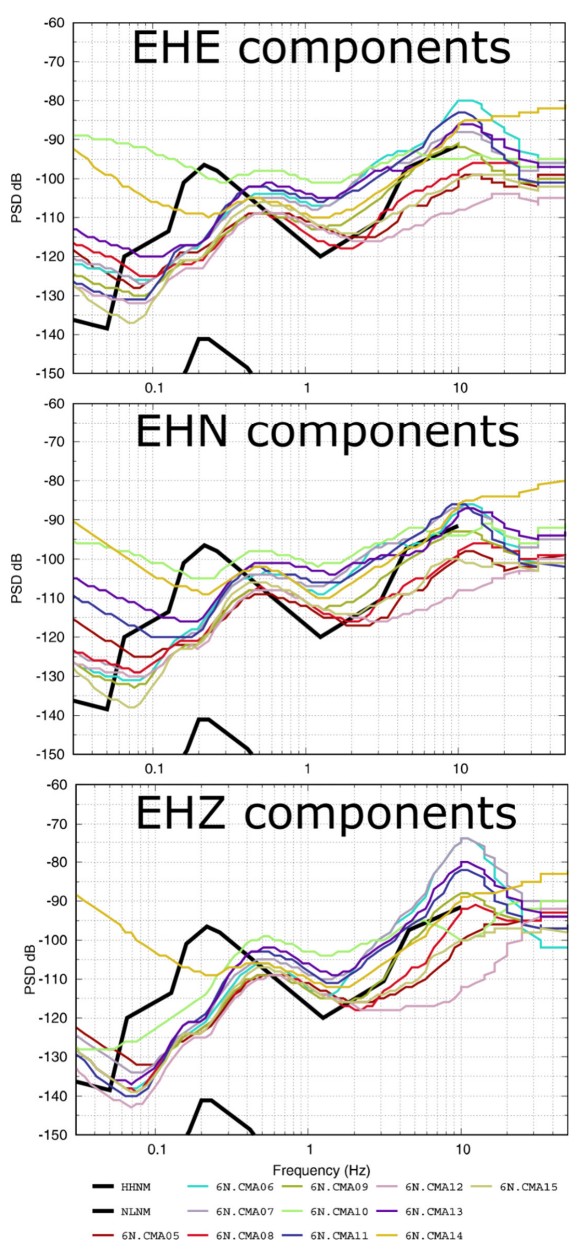

**Figure 9.** 90th percentile curves of PSD computed for all stations on the three components of motion.

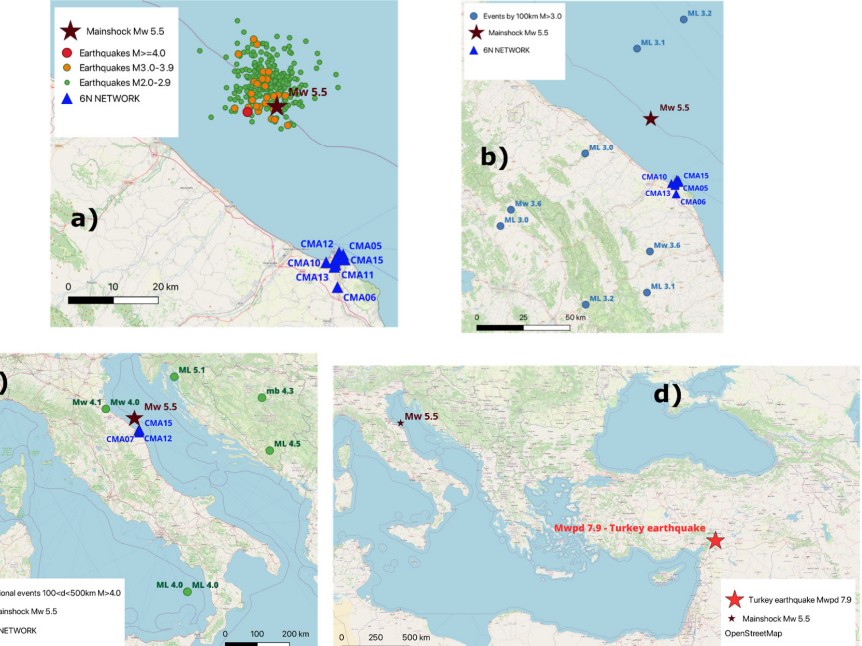

**Figure 10.** Seismicity during the operation of the 6N network: a) Costa Marchigiana-Pesarese seismic sequence; b) Events of other italian seismic sources within 100km from Ancona; c) Regional events; d) Teleseismic Turkey event. © OpenStreetMap contributors 2024. Distributed under the Open Data Commons Open Database License (ODbL) v1.0.



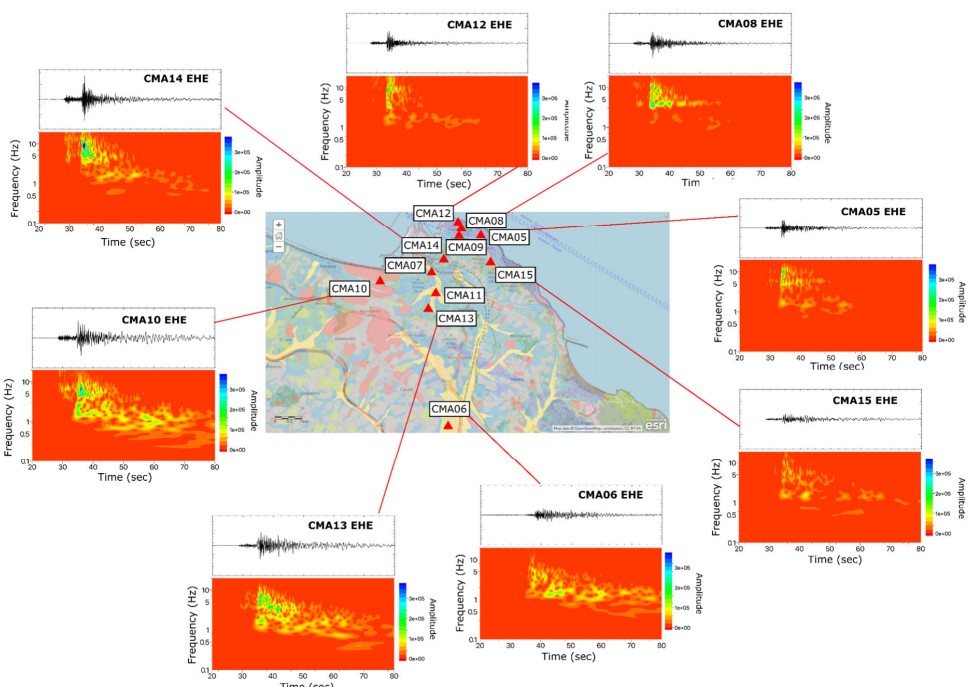

**Figure 11.** Time series and spectrograms of the M$_W$ 3.9 earthquake (EHE components) occurred the 8th of December, 2022 at 07:08:18 UTC for some stations of the 6N network.

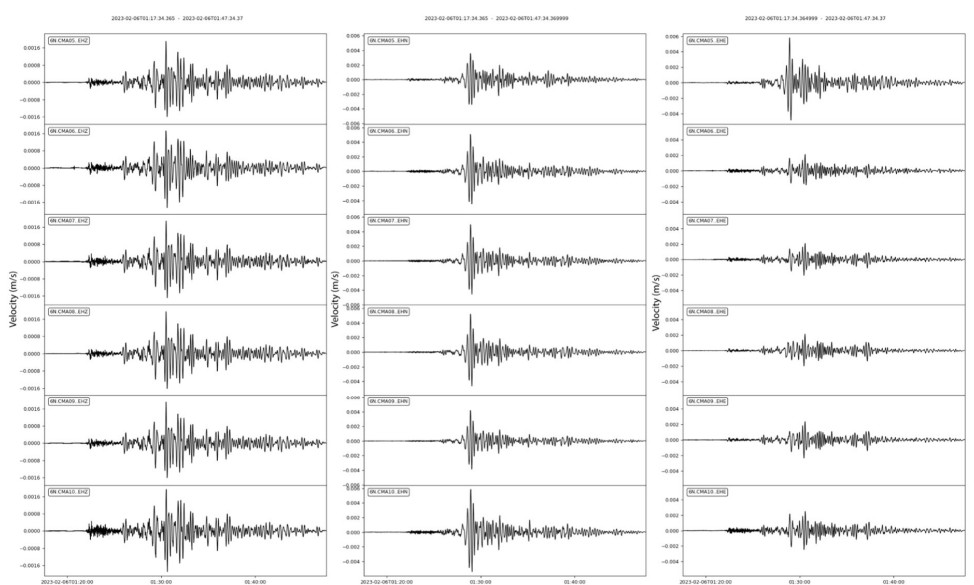


**Figure 12.** Seismic traces of the Mwpd 7.9 Turkish earthquake occurred the 6th of February 2022 (01:17 UTC) recorded by the real-time 6N EMERSITO stations.

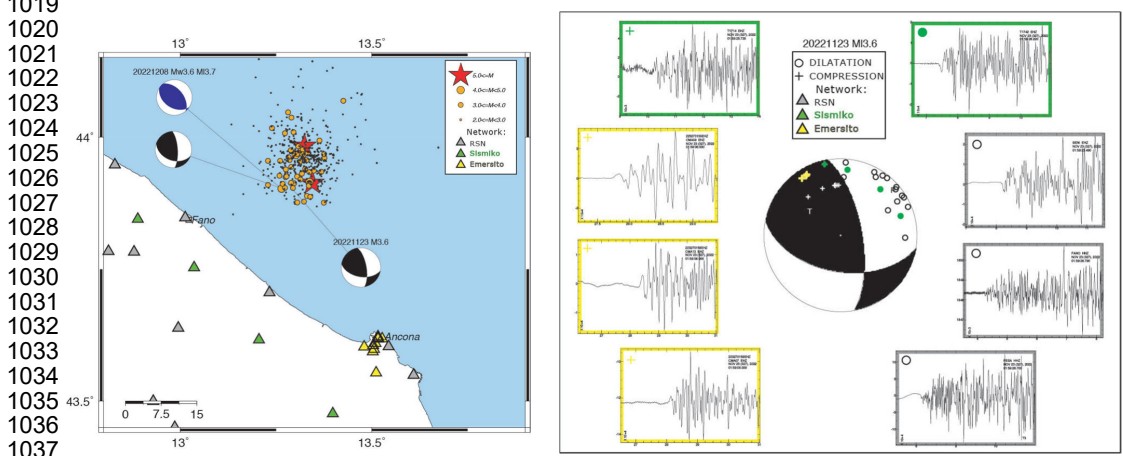

**Figure 13.** Left: Fault-Plane fit (FPFIT) focal solutions (black) for the earthquakes reported in Table 3. For the second event (id #33589291) the available Time Domain Moment Tensor (TDMT) is also show in (blue); circles are M>=2.0 earthquakes of the seismic sequence (see the insert for the different sizes and the correspondence with difference magnitude), and red stars are M>=5.0 earthquakes.

Right: distribution of polarities, up and down, for the first event in Table 3 (id #33466171). Seismograms recorded by Y1 (green boxes), IV (grey boxes) and 6N (yellow boxes) networks are also shown.



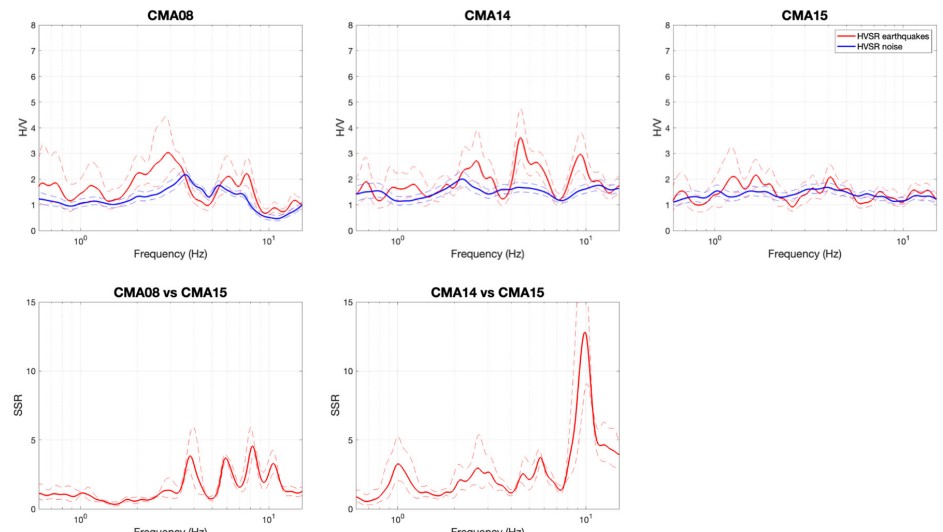


**Figure 14.** Top: HVNSR (blue lines) and HVSR (red lines) from HVNEA for CMA08, CMA14 and CMA15 stations. Bottom: SSR for CMA08 and CMA14 stations (red lines). For all plots, the solid lines are the averages, the dotted lines the average minus and plus one standard deviation.

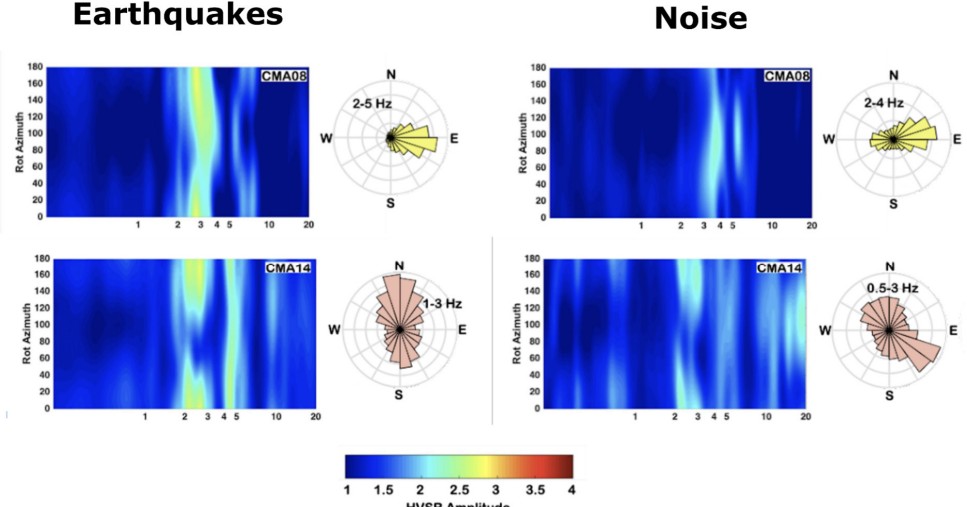

**Figure 15.** Directional amplification at two exemplificative stations: CMA08 (top) and CMA14 (bottom), by using seismic events (left-hand side) and ambient noise recordings (right-hand side). Rotated HVSR and HVNSR are graphed as contour plots, where the color scale is related to the amplitude level, the x-axis represents frequency, the y-axis the rotation angle (0° and 180° corresponding to N-S direction, 90° to EW direction). The time-domain polarization analysis is summarized by means of circular histogram diagrams representing the polarization angle in the horizontal plane, obtained from filtered signals in the frequency band indicated in the rose diagram.





**Figure 16.** Summary of the HVSR analyses performed on ambient noise and earthquake recordings, by using only the mean of the two horizontal components and by calculating rotated components. The circle dimension plotted above each station is related to the HVSR $A_0$ value, while its colour indicates the $F_0$ value. In case of directional amplification, we also add rose diagrams (gray and white colours are related to results retrieved using earthquakes and ambient noise, respectively). The results are superimposed to the 1:10.000 geological map.

© OpenStreetMap contributors 2024. Distributed under the Open Data Commons Open Database License (ODbL) v1.0.