# Peer review of "Seismic survey in urban area: the activities of the EMERSITO INGV emergency group in Ancona (Italy) following the 2022 $M_W$ 5.5 Costa Marchigiana-Pesarese earthquake"

_Earth System Science Data, 2024_

## Author Response (AR1)

**Reviewer 1**:

The manuscripts reports the activities of the INGV emergency group EMERSITO following the seismic sequence generated by the Mw 5.5, 2022 earthquake that occurred along the Adriatic coast. The manuscript provides details on the installation of a temporary network of 11 seismic stations to monitor the evolution of the aftershocks. As the aim of the deployment was to collect seismic information for future microzonation activities, the authors provide information on the geological setting of the monitored area and on the earthquake data collected. The latter are used to characterize the spatial variability of the site amplification effects considering different signals (i.e. earthquake data and noise) and techniques (i.e. standard spectral ratio and horizontal/vertical spectral ratio), and also to investigate azimuthal dependent features of the site responses. The authors also showed the contribution of the temporary networks in improving the location of events during the 2022 seismic sequence and their focal mechanisms.

In terms of data sharing, the manuscript indicates how to access the recorded seismic data, stored in EIDA and freely available, and provides for each station a pdf file with a summary of the location of each station, maps of the location and figures of the results of the site amplification investigations.

Comments:

As the manuscript reports in detail on the temporary survey, the data collection and the results obtained in terms of site amplifications, I have no further requirements regarding the text. My only suggestion is that the results of the site amplifications should also be made available in electronic format. The summary station reports (i.e., the pdf files) are useful, but I think the role of ESSD is to provide a data product in a form that other authors can use independently. Here the raw data are shared via EIDA, and this is an important point, but it is independent of the manuscript (they would be shared anyway). The main contribution of the manuscript is the detailed site amplification information obtained for each station, which could be used in future microzonation studies or for other purposes.

Therefore, I would strongly encourage the authors to create and share an archive with folders containing the results for each station. Within each folder, the authors could store the results of the H/V and SSR analysis in electronic format (e.g. frequency dependent means and standard deviations of H/V and SSR; distribution of azimuthal dependent results, etc.), allowing other users to use the results of the present manuscript in their own research.

***Authors***: *First of all, we thank the reviewer for the comments and suggestions to improve the manuscript. Following his points we decided to make freely available the results of the preliminary analyses and summarized in the station reports. For most of the analyses (HVNSR, HVSR, SSR, rotated HVNSR and rotated HVSR), we uploaded the results in the Zenodo repository and referenced them in the text. It was not possible to share the rose diagrams in electronic format because unfortunately the output format of this kind of analysis is not very convenient, being adapted to produce the graphic only. Anyhow, similar information can be extracted from the rotated H/V.*

Minor details:

1) In the description of how to access the raw data stored in EIDA using WebDc3, the authors could also mention that waveforms and metadata can be easily retrieved and downloaded from EIDA programmatically using FDSN web services.

***Authors****: According to this request, we modified the text in the  "Details on dataset access" section, adding the INGV Web Services based on FDSNWS specifications as an alternative method to retrieve and download the waveforms and metadata from EIDA.*

2) In Table 1, I would add a column summarizing the type of installation (e.g., free field, ground floor of a single store building; basement of a multi-store building, etc.).
***Authors****: We think the reviewer was talking about Table 2 and not Table 1 that refer to stations belonging to the permanent networks of INGV and RAN. Therefore, we added a column in Table 2 that indicates the type of installation of our 6N temporary network. Anyhow, to fulfill any request about the stations of permanent stations, in the caption of Table 1 we added a sentence saying that "more info about stations of IV and IT networks can be found on the ITalian ACcelerometric Archive (ITACA) and on the Site characterization of the permanent stations database (CRISP)" and added the reference of ITACA and CRISP.*

**Reviewer 2**:
The paper details the activities of the EMERSITO INGV emergency task force (devoted to site effect and microzonation studies during significant seismic crises) in Italy in Ancona, Italy, following the 2022 MW 5.5 Costa Marchigiana-Pesarese earthquake. The group deployed a temporary seismic network in Ancona to study site effects and seismic microzonation. The network, operational from November 2022 to February 2023, collected data to analyze local seismic responses. The study included the installation of seismic stations, data collection, and preliminary analyses using various techniques like HVNSR, HVSR, and SSR. The findings highlighted the geological and seismic characteristics of Ancona, contributing to improved earthquake parameter estimates and understanding of local seismic responses. The data is available for further research through the EIDA database.

The activity of the INGV Task forces, as described in the paper, demonstrates great professionalism and good organization, all with the aim of minimizing the harmful effects of earthquakes and increasing the safety of citizens. The paper provides some interesting (and thought-provoking) conclusion points.

The paper reports on the time-limited research, the data collection and the results obtained in terms of amplification. One of my comments is that the results of the amplification (which is the main contribution of the paper) should be made available in electronic form. Also, further research should densify the instrument network to make the potential microzonation more reliable and include a larger number of earthquakes in the analysis, as the authors themselves suggest.

*Authors: Many thanks to the reviewer for all the comments. The manuscript has been modified following the points raised by the reviewer. First, we decided to make available the results of the preliminary analysis and summarized in the station reports. For most of the analyses (HVNSR, HVSR, SSR, rotated HVNSR and rotated HVSR), we uploaded the results in the Zenodo repository and referenced them in the text. It was not possible to share the rose diagrams in electronic format because unfortunately the output format of this kind of analysis is not very convenient, being adapted to produce the graphic only. Anyhow similar information can be extracted from the rotated H/V.*

*Hopefully, further research studies would interest Ancona in the future and for sure our dataset requires to be analysed deeply.*

Some minor comments:
Fig. 2 - What intensity or degree of damage do the damage marked in orange correspond to, and what does the brown (dark-red?) damage correspond to?

*Authors: we reviewed the information obtained by the Fire Brigade, and we realized that our description in the text was not very accurate. We are sorry for the inconvenience. The Fire Brigade did not estimate a level of damage but inspected buildings and outdoor public areas. Then, they indicated with the different colours:*
*a) the partial banning of buildings*
*b) the complete banning of buildings*
*c) the banning of outdoor public areas*
*According to this information, we corrected Fig. 2, the text in the manuscript and the caption of Fig. 2.*

Table 1 - Indicate in the title of the table to which earthquake the indicated values (epicentral distance and PGA) refer.
In Table indicate type of the sensor and its location.
***Authors****: we added in the caption of Table 1 the earthquake to which the indicated values (epicentral distance and PGA) refer to.*
*In Table 1 we added two columns indicating the latitude and the longitude of stations of the permanent networks IV and IT, and another column indicating the type of sensor. This info as well as others of interest can be found on the ITalian ACcelerometric Archive (ITACA) and on the Site characterization of the permanent stations database (CRISP).*

Show the geotectonic figure of the area (in Fig. 1), wider than in Fig. 4.
***Authors****: For the right-hand part of Fig. 1 we added the geological map in 1:500000 scale and the individual seismogenic sources as reported by the DISS Working Group (2021). Of course, we also added a legend for the geology. Hopefully, this new version of Fig. 1 would fulfill the request of the reviewer about the geotectonics of the area.*

Fig. 5 - Add a better quality legend, especially with regard to geophysical measurements/research at the locations marked with colored dots.
***Authors****: This figure was only intended to show a generic example of the Web-GIS project. Anyhow, we added a better quality legend indicating the geophysical measurements/research at the locations marked with colored dots.*

Fig. 6 - Labeling of isolines with numbers for easier reference.
***Authors****: Done, we added labels of isolines with numbers for easier reference*

Table 2 - What type of instruments/networks are involved? SM or WM?
***Authors****: In the caption of Table 2 we added a sentence saying that "the 6N seismic network [is] equipped with both accelerometric and velocimetric sensors."*

**List of all relevant changes made in the manuscript**

1) Caption of Table 1, added two sentences:
   "November 9th, 2022, MW 5.5 earthquake: "
   "More info about stations of IV and IT networks can be found on the ITalian ACcelerometric Archive (ITACA[17]) and on the Site characterization of the permanent stations database (CRISP[18])""

   The new caption is then:
   "November 9th, 2022, MW 5.5 earthquake: PGA recorded by some stations of the two permanent networks in Italy, IV (https://doi.org/10.13127/SD/X0FXnH7QfY) and IT (https://doi.org/10.7914/SN/IT), ordered by epicentral distance. The two stations in Ancona are highlighted in bold. More info about stations of IV and IT networks can be found on the ITalian ACcelerometric Archive (ITACA[17]) and on the Site characterization of the permanent stations database (CRISP[18])."

2) Added tree columns to Table 1:
   LAT, LON and Sensor Type
   The new Table 1 is:

| Network | Station | Locality | Epicentral distance (km) | Horizontal PGA (cm/s²) | LAT (Decimal degrees) | LON (Decimal degrees) | Sensor type |
|---|---|---|---|---|---|---|---|
| IV | COR1 | Corinaldo | 49.3 | 31.610 | 43.6318 | 13.0003 | Velocimeter + accelerometer |
| **IT** | **ANB** | **Ancona** | **48.8** | **166.424** | **43.592** | **13.507** | **Accelerometer** |
| IV | FCOR | Fonte Corniale | 48.6 | 21.796 | 43.7691 | 12.8145 | Accelerometer |
| **IV** | **PCRO** | **Ancona** | **47.9** | **197.842** | **43.6076** | **13.5323** | **Accelerometer** |
| IT | CTL | Cattolica | 47.3 | 31.749 | 43.955 | 12.736 | Accelerometer |
| IV | CRTC | Cartoceto | 44.2 | 22.409 | 43.7684 | 12.8830 | Velocimeter + accelerometer |
| IV | SENI | Senigallia | 34.6 | 139.209 | 43.7052 | 13.2331 | Velocimeter + accelerometer |
| IV | FANO | Fano | 30.5 | 52.613 | 43.8434 | 13.0183 | Accelerometer |

3) Section 2.2 EMERSITO INGV intervention - Changed a sentence close to Line 228-229

   "Afterwards, the Fire Brigade performed a detailed survey for all buildings,

distinguishing the levels of damage in the city"

with

"Afterwards, the Fire Brigade performed a detailed survey for all buildings and public areas, distinguishing the partial and complete banning of buildings and the banning of outdoor public areas"

4) Caption of Table 2, added a sentence: ", equipped with both accelerometric and velocimetric sensors."

The new caption is then:
"List of the sites of the 6N seismic network, equipped with both accelerometric and velocimetric sensors. The dismissing date of the stations was 24th of February 2023."

5) Added one column to Table 2 named "Type of installation"
The new Table 2 is:

| Name | Location | Lat | Lon | Installation date | Acquisition mode | Type of installation |
|---|---|---|---|---|---|---|
| CMA05 | Piaget School | 43.618437 | 13.52708 | 2022-11-15 10:40 | Real Time | basement of a multistore building |
| CMA06 | Paolinelli Sports Center, in the hamlet of Baraccola | 43.553738 | 13.511387 | 2022-11-15 11:32 | Real Time | free field |
| CMA07 | Salesian Oratory | 43.605702 | 13.503745 | 2022-11-13 18:03 | Real Time | ground floor of a multistore building |
| CMA08 | Economics University | 43.620228 | 13.516387 | 2022-11-14 15:12 | Real Time | basement of a multistore building |
| CMA09 | Church of Saints Cosma and Damiano | 43.618237 | 13.515918 | 2022-11-13 11:12 | Real Time | basement of a multistore building |
| CMA10 | Via della Grotta (landslide) | 43.603008 | 13.480115 | 2022-11-14 11:18 | Real Time | free field |
| CMA11 | Navy | 43.598542 | 13.506017 | 2022-11-14 16:05 | Stand Alone | ground floor of a 1-store building |
| CMA12 | Cardeto park (lighthouse) | 43.622585 | 13.51589 | 2022-11-15 10:40 | Stand Alone | ground floor of a 1-store building |
| CMA13 | Via Barilatti | 43.593848 | 13.502273 | 2022-11-15 13:33 | Stand Alone | basement of a multistore building |
| CMA14 | Raffaello Palace | 43.609948 | 13.509390 | 2022-11-15 16:07 | Stand Alone | basement of a multistore building |

| CMA15 | Palascherma | 43.608372 | 13.531515 | 2022-11-15 16:08 | Stand Alone | ground floor of a multistore building |
|---|---|---|---|---|---|---|

6) Section 4.2.1 - Added a reference for HVSR technique. This reference is: Lermo and Chávez-García, 1993
The complete reference has been added to the References list.

7) Section 4.3 Summary results, added sentences to say that most of the results can be downloaded from Zenodo.
The complete added paragraph is:

"Moreover, the results can be can be accessed and downloaded in electronic format at Zenodo under:
1) HVNSR curves: 10.5281/zenodo.14671630 (Cara and Famiani, 2025)
2) HVSR curves: 10.5281/zenodo.14672463 (Cara and Famiani, 2025)
3) SSR curves: 10.5281/zenodo.14672942 (Cara and Famiani, 2025)
4) Rotated HVNSR curves: 10.5281/zenodo.14700834 (Pischiutta et al., 2025)
5) Rotated HVSR curves: 10.5281/zenodo.14701170 (Pischiutta et al., 2025)."

8) "Details on dataset access" has been modified as follow:

The dataset uploaded to EIDA can be requested in two ways:

1) Using the Orfeus Data Center WebDC3 Web InterfaceThe best way for downloading data from EIDA is to use the Orfeus Data Center WebDC3 Web Interface

[revised manuscript text omitted]

13) According to the reviewers' comments we changed Figure 5, adding more readable legend.

The new Figure 5 is:

[Figure]

14) According to the reviewers' comments we changed Figure 6, adding the numbers close to the isolines..

The new Figure 6 is:

[Figure]

15) We uploaded once again the supplementary files because the name of one of the zip files was incorrect. The content was right

---

## Author Response (AR2)

**Point 1:**

*Dear Daniela Famiani and co-authors,*
*many thanks for the revision of your manuscript. The referees and myself are happy with the result and I am accepting the manuscript for final publication after making another slight addition: As you have published the results (HVNSR, HVSR and SSR curves) newly on Zenodo, I suggest that you add these new data publications also to the data availability statement. This section shall provide the overview on all data and not - as currently - "only" the link to the EMERSITO seismic network*

*Many thanks and best regards,*
*Kirsten Elger*

**Authors**: *Thank you for accepting the manuscript for final publication. Following your request, we have added the list of results published on Zenodo to the "Data Availability" section.*
*The review also helped us realize that the section number, as well as the subsequent ones, was incorrect. It was labeled as number 6, but it should have been number 5. We have corrected it accordingly, along with the numbering of the following section.*
*The section "Data Availability" is now as follow (additions are highlighted in green):*

**5 Data Availability**

Data described in this manuscript can be accessed under 10.13127/sd/qctgd6c-3a (EMERSITO Working Group, 2024).
Moreover, the results can be can be accessed and downloaded in electronic format at Zenodo under:
   1) HVNSR curves: 10.5281/zenodo.14704661 (Cara and Famiani, 2025)
   2) HVSR curves: 10.5281/zenodo.14672464 (Cara and Famiani, 2025)
   3) SSR curves: 10.5281/zenodo.14672943 (Cara and Famiani, 2025)
   4) Rotated HVNSR curves: 10.5281/zenodo.14700835 (Pischiutta et al., 2025)
Rotated HVSR curves: 10.5281/zenodo.14701171 (Pischiutta et al., 2025).

*Consequently, we also had to modify another part of the text that included the same list. This concerns the first part of section 4.3 'Summary of results,' which now reads as follows (additions are highlighted in green):*

**4.3 Summary results**

This subsection illustrates the results of the techniques described in the previous sections, by using three selected stations as representative of the network: CMA08, CMA14 and CMA15. The results for all the stations of the 6N network are given as synthetic sheets and collected in the supplementary material (Figures from S3 to S13). Moreover, the results have been uploaded to the Zenodo repository (see Section 5 for details).

**Point 2**:

*Checking your paper, I noticed that your tables contain coloured cells. Please note that this will not be possible in the final revised version of the paper due to HTML conversion of the paper. When revising the final version, you can use footnotes or italic/bold font. For now, the process will continue, but please note that the final version cannot be published by using coloured tables.*

**Authors***: We have replaced all colored cells in the tables with non-colored ones.*